



# 1 Importance of multiple sources of iron for the upper ocean
# 2 biogeochemistry over the northern Indian Ocean

Priyanka Banerjee[1]
[1]Divecha Centre for Climate Change, Indian Institute of Science, Bangalore, India.
*Correspondence to*: Priyanka Banerjee (pbanerjee@iisc.ac.in)
**Abstract**
Although the northern Indian Ocean (IO) is globally one of the most productive regions and receives dissolved
iron (DFe) from multiple sources, there is no comprehensive understanding of how these different sources of DFe
can impact upper ocean biogeochemical dynamics. Using an Earth system model with an ocean biogeochemistry
component this study shows that atmospheric deposition is the most important source of DFe to the upper 100 m
of the northern IO, contributing more than 50% of the annual DFe concentration. Sedimentary sources are locally
important in the vicinity of the continental shelves and over the southern tropical IO, away from high atmospheric
depositions. While atmospheric deposition contributes to more than 10% (35%) to 0-100 m (surface level)
chlorophyll concentrations over large parts of the northern IO, sedimentary sources have similar contribution to
chlorophyll concentrations over the southern tropical IO. Such increases in chlorophyll are primarily driven by an
increase in diatom population over most of the northern IO. The regions that are susceptible to chlorophyll
enhancement following external DFe additions are where low levels of background DFe and high background
$NO_3$:DFe values are observed. Analysis of DFe budget over selected biophysical regimes over the northern IO
points to vertical mixing as most important for DFe supply, while the importance of advection (horizontal and
vertical) varies seasonally. Apart from removal of surface DFe by phytoplankton uptake, subsurface balance
between DFe scavenging and regeneration is crucial in replenishing DFe pool to be made available to surface
layer by physical processes.
**1    Introduction**
Iron is an essential micronutrient for primary producers in the ocean due to the catalytic role of iron in
photosynthesis, respiration, and nitrogen fixation (Geider & La Roche, 1994; Raven, 1988). Although iron is one
of the most abundant elements in the Earth's crust (McLennan, 2001), its low solubility (Sholkovitz et al., 2012)
coupled with an intricate balance between complexation by ligands and high scavenging tendency does not make
it readily bioavailable (Boyd & Ellwood, 2010). It has been estimated that iron availability limits primary
productivity in as much as ~30% of the global oceans, which results in accumulation of unutilized macronutrients
like nitrate and phosphate (Moore et al., 2013a). Even in regions experiencing nitrate limitation of productivity,
nitrogen fixation is controlled by the supply of iron (e.g., Mills et al., 2004; Moore et al., 2009; Schlosser et al.,
2014). Several iron addition experiments performed in the open oceans have demonstrated its significance in
regulating phytoplankton growth and drawdown of atmospheric $CO_2$ (e.g., Blain et al., 2007; Boyd et al., 2007;
Coale et al., 1996; de Baar et al., 2005; Pollard et al., 2009).



The main external sources of dissolved iron (DFe) to the world oceans are atmospheric depositions (e.g., Conway
et al., 2014; Jickells et al., 2005), continental sediments (Elrod et al., 2004; Johnson et al., 1999), river inputs (e.g.,
Buck et al., 2007; Canfield, 1997), sea ice (Sedwick & DiTullio, 1997; Wang et al., 2014) and iron seeping from
hydrothermal vents (e.g., Nishioka et al., 2013; Tagliabue et al., 2010). Most ocean biogeochemistry models
simulating the iron cycle estimate dust (1.4-32.7 Gmol yr$^{-1}$) or sedimentary sources (0.6–194 Gmol yr$^{-1}$) to have
the highest contribution to ocean DFe inventory (Tagliabue et al., 2016). However, many of these models do not
include hydrothermal sources of DFe. Numerical modelling using dust, sedimentary and hydrothermal sources of
DFe have shown that while ocean column DFe inventory is most sensitive to sedimentary and hydrothermal DFe,
atmospheric and sedimentary sources of DFe have the largest impact on atmospheric carbon dioxide (Tagliabue
et al., 2014). This is because hydrothermal vents can only impact productivity where these vents are located at
shallow depths, while atmospheric and sedimentary DFe can impact productivity over both the open and coastal
ocean regions. However, with availability of more *in situ* DFe measurements, the relative importance of different
sources of DFe are being re-examined at global as well as regional scales.
The northern Indian Ocean (IO) is one of the most productive regions of the global oceans, contributing high
levels of organic carbon fluxes to the deeper ocean (e.g., Barber et al, 2001; Madhupratap et al., 2003; Rixen et
al., 2019). The monsoonal winds drive phytoplankton blooms over different regions of the northern IO, arising
from distinct physical mechanisms in different seasons. These mechanisms include blooms due to coastal and
open ocean upwelling, advection of nutrients by ocean currents, and mixed layer deepening by winter convection.
Episodic blooms are also triggered by passage of cyclones (Kuttippurath et al., 2021) and mesoscale eddies
(Prasanna Kumar et al., 2004; Vidya & Prasanna Kumar, 2013). The region hosts one of the most intense oxygen
minimum zones of the world oceans (Schmidtko et al., 2017) and is globally one of the major denitrification sites
(e.g., Morrison et al., 1999; Bianchi et al., 2012). Several water column measurements have shown that the primary
limiting nutrient over the northern IO is reactive nitrogen with possible colimitation by silicate (Końe et al., 2009;
Moore et al., 2013a; Morrison et al., 1998). In recent years, a few studies using ocean biogeochemistry models
have also pointed to possible iron limitation of phytoplankton blooms during southwest monsoon months (June-
September), especially over upwelling regions of the western Arabian Sea (AS), which is the north-western part
of the IO (Końe et al., 2009; Wiggert et al., 2007). These findings on the role of iron limitation have also been
supported by incubation experiments over the AS during the late southwest monsoon, which have noted
chlorophyll enhancements following iron enrichments (Moffett et al., 2015). Furthermore, *in situ* measurements
during the late southwest monsoon have revealed complete drawdowns of silicate, owing to its high utilization
under iron limitation, as well as high nitrate-to-iron ratios over the western AS (Naqvi et al., 2010). Nutrient
enrichment experiments over the central AS during northeast monsoon months (December-March) have also
revealed signatures of iron and nitrate colimitation, with addition of these two nutrients supporting increases in
diatoms and coccolithophores (Takeda et al., 1995). Colimitation by nitrogen, phosphorus and iron has been
identified over the southern Bay of Bengal (BoB, the north-eastern part of the IO) and the eastern equatorial IO
(Twining et al., 2019). Thus, availability of iron can have major impacts on availability of other macronutrients
and productivity, which can in turn impact denitrification and mid-depth oxygen levels in this region by
modulating fluxes of sinking organic matters.



In general, there is a reduction in surface DFe concentrations over the northern IO from north to south. Systematic DFe measurements, encompassing all seasons over the AS, conducted during the Joint Global Ocean Flux Study (JGOFS) of the 1990s showed DFe concentrations often exceeding 1 nM, especially during the southwest monsoon (Measures & Vink, 1999). Subsequent measurements revealed lower levels of DFe with surface values ranging between 0.2-1.2 nM over the AS and between 0.2-0.5 nM over the BoB (Chinni et al., 2019; Chinni & Singh, 2022; Grand et al., 2015; Moffett et al., 2015; Vu & Sohrin, 2013). These values are generally higher than most of the open ocean regions. In contrast, southwards of the equatorial IO have surface DFe values generally less than 0.2 nM (e.g., Chinni et al., 2019; Grand et al., 2015; Twining et al 2019; Vu & Sohrin, 2013). The oxygen minimum zone, located to the north of the equator between depths of 150-1000 m, has elevated levels of DFe (>1 nM), possibly due to DFe transport from reducing shelf sediments and remineralization of sinking organic matter (Moffett et al., 2007).

The overall high values of DFe over the northern IO can stem from multiple external sources of DFe identified within this region: atmospheric aerosol inputs (dust and black carbon) from South and Southwest Asia (Banerjee et al., 2019; Srinivas et al., 2012), continental shelf sediments, high river discharge, especially, over the BoB (e.g., Chinni et al., 2019; Grand et al., 2015) and hydrothermal vents from the Central Indian Ridge that mainly impact DFe levels at depths of around 3000 m (Nishioka et al., 2013). The importance of episodic dust depositions in alleviating iron limitations of primary productivity over the central AS has been identified, during the northeast monsoon when a deeper ferricline compared to the nitracline yields a high nitrate-to-iron ratio (Banerjee and Kumar, 2014). Additionally, modelling studies over the AS have demonstrated that DFe derived from dust deposition can support about half of the observed primary productivity and a large fraction of nitrogen fixation (Guieu et al., 2019). Centennial-scale model simulations over the IO have revealed that changes in phytoplankton community structure have resulted in increased (reduced) carbon uptake over the eastern (western) IO in response to increased anthropogenic DFe deposition in the present day compared to pre-industrial levels (Pham & Ito, 2021). Yet another challenge is that, away from regions with high aerosol loading, other sources of DFe can become important in supporting ocean productivity and controlling patterns of nutrient limitations. Such understanding of relative roles of different sources of DFe in controlling the biogeochemical dynamics of the northern IO remains unexplored. This is important considering the multiple sources of DFe over the northern IO. To this end, the present study uses a suite of simulations from a state-of-the art Earth system model with an iron cycle in its ocean biogeochemistry component to explore the relative contribution of different sources of DFe to phytoplankton blooms and impacts on nutrient availability over the upper 100 m of the northern IO. Furthermore, DFe budget has been analysed over the upper ocean for varied biophysical regimes in this region to identify how different sources of DFe can impact the total DFe budget.

## 2    Data and model

The study uses satellite and reanalysis products, ocean observation data, and an Earth system model to assess contributions of different sources of DFe to phytoplankton blooms over the northern IO. For the present study, the northern IO is considered to encompass 30ºN–20ºS latitude, 40º–105ºE longitude. Thus, the tropical part of the southern IO is also included. Only the open ocean regions, having bottom depth greater than 1000 m, are studied here. The four seasons referred to in this study are defined as: the northeast monsoon: December-March;



spring intermonsoon: April-May; southwest monsoon: June-September; and fall intermonsoon: October-
November.

### 2.1 Model

This study uses the ocean component Parallel Ocean Program version 2 (POP2) (Smith et al., 2010) embedded in
the Community Earth System Model (CESM) version 2.1. This version of CESM incorporates several
improvements over previous versions of the model (Danabasoglu et al., 2020). The POP2 model is a level-
coordinate model having Arakawa B-grid in the horizontal with North Pole displaced over Greenland. The vertical
resolution is 10 m for the upper 160 m and decreases with depth to 250 m in the bottom. The horizontal resolution
is nominally $1°$ with meridional resolution increasing to $0.27°$ near the equator (Danabasoglu et al., 2012),
implying that mesoscale eddies are not resolved. Momentum advection is based on a second-order central
advection scheme while tracer advection relies on a third-order upwind advection scheme. Vertical ocean mixing
is parameterized using the non-local K-Profile parameterization (Large et al., 1994), which is incorporated into
CESM2.1 via the Community Ocean Vertical Mixing (CVMix) framework. Horizontal mixing is parameterized
using the Gent and Williams (1990) scheme, which includes eddy-induced velocity in addition to diffusion of
tracers along isopycnals. Macronutrients and oxygen are initialized from World Ocean Atlas 2013 version 2
dataset (Garcia et al., 2014a, b) and alkalinity is initialized using GLobal Ocean Data Analysis Project
(GLODAPv2; Olsen et al., 2016).
The biogeochemistry component of POP2 is implemented using Marine Biogeochemistry Library (MARBL),
which is the most updated version of the previously implemented Biogeochemistry Elemental Cycle (BEC) model
(Long et al., 2021). The model includes key limiting nutrients (N, P, Si, Fe), three types of explicit phytoplankton
functional groups (diatoms, diazotrophs and nano/picophytoplankton), one implicit calcifier group, and one
zooplankton type. The C:N ratio for nutrient assimilation is fixed at 117:16 (Anderson and Sarmiento,1994),
whereas P:C, Fe:C, Si:C and chlorophyll:C ratios are allowed to vary based on ambient nutrient concentrations.
The Fe:C ratio is allowed to change within a fixed range based on phytoplankton growth terms, loss terms, and
the iron uptake half-saturation constant for different phytoplankton groups (Moore et al., 2004). For each of the 3
phytoplankton groups the minimum allowed Fe:C ratio is 2.5 µmol mol⁻¹. The maximum allowed Fe:C ratio is 30
µmol mol⁻¹ for diatoms and small phytoplankton, and 60 µmol mol⁻¹ for diazotrophs due to their higher demand
for iron. The zooplankton Fe:C ratio is fixed at 3.0 µmol mol⁻¹. Individual nutrient limitation for phytoplankton is
assessed based on Michaelis-Menten nutrient uptake kinetics, which is a function of the specific nutrient
concentration and nutrient uptake half-saturation coefficient. The half-saturation coefficient is nutrient-specific
and phytoplankton-group specific. Nutrient limitation terms vary from 0 to 1, with 0 being the most limiting
nutrient. Multiple nutrient limitation follows Liebig's law of minimum, so that the nutrient limitation term with
minimum value limits phytoplankton growth rate (Long et al., 2021). Loss of phytoplankton in MARBL is
accounted for by grazing, mortality, and aggregation of sinking flocculants.
The main DFe sources considered in MARBL are atmospheric depositions, shelf sediments, riverine inputs, and
hydrothermal vents. Globally, these sources of DFe account for 13.62 Gmol yr⁻¹, 19.68 Gmol yr⁻¹, 0.37 Gmol yr⁻
¹, and 4.91 Gmol yr⁻¹, respectively (Long et al., 2021). Atmospheric sources of DFe are from dust and black carbon
depositions obtained from a fully coupled CESM2 simulation in hindcast mode at nominal $1°$ spatial resolution as



a part of the Coupled Model Intercomparison Phase 6 (CMIP6) contribution. Dust emissions and
transport/deposition are calculated, respectively, using the Community Land Model version 5 (CLM5) and
Community Atmosphere model version 6 (CAM6) in Whole Atmosphere Community Climate Model (WACCM)
configuration. The newly included Modal Aerosol Module version 4 (MAM4) in CAM6 includes dust in the
accumulation and coarse modes. Black carbon is emitted in the primary mode and transferred to accumulation
mode via aging (Liu et al., 2016). Monthly climatology of dust and black carbon for the year 2000 is used in
repeating mode. About 3.5% of dust is assumed to be iron with the solubility of iron depending on the ratio
between coarse and fine dust fluxes. This accounts for increasing iron solubility with increasing distance from
dust source regions. A constant solubility of 6% is assigned to iron derived from black carbon aerosols.
Sedimentary iron supply is based on sub-grid scale bathymetry that depends on two factors: firstly, for reducing
sediments, it is proportional to particulate organic carbon fluxes in regions where these fluxes are larger than 3 g
C m$^{-2}$ yr$^{-1}$; secondly, in oxic sediments, it depends on constant low background fluxes and bottom current velocity,
which accounts for sediment resuspension. As a result, the main sources of sedimentary DFe are along continental
shelves and productive margins, with little contribution coming from the deep ocean. For the river source of DFe,
discharge data for the year 2000 from Global Nutrient Export from WaterSheds (GlobalNEWS, Mayorga et al.,
2010) is combined with constant DFe concentration of 10 nM. For hydrothermal vents, a constant flux of iron
from the grid boxes containing vents is applied so that the total hydrothermal vent iron flux is equal to
approximately 5.0 Gmol yr$^{-1}$.
Iron input to oceans is balanced by losses from biological uptake and scavenging. Loss of iron from the biological
pool occurs through mortality and grazing upon phytoplankton by zooplankton as well as higher trophic grazing
on zooplankton. In CESM, scavenging increases non-linearly with DFe concentration. The scavenging rate
depends on the total sinking fluxes of particulate organic carbon, biogenic silica, calcium carbonate and dust,
which strongly influence DFe in excess of ligand concentrations (Moore and Braucher, 2008). Scavenged iron
enters the particulate iron pool, while iron released from grazing and mortality of autotrophs and zooplankton also
contributes to the particulate iron pool depending on species-specific Fe:C ratios. Remineralization of particulate
iron at depth is parameterized as a function of the particulate organic carbon flux. Desorption of iron contributes
to the remineralized iron pool and is calculated using a constant desorption rate for scavenged iron. In addition,
there is slow dissolution of "hard" dust fraction (~98% of total dust) with depth such that ~0.3% of dust will
dissolve over 4000 m (Armstrong et al., 2002; Moore et al., 2004). For the remainder of the 2% "soft" dust,
remineralization takes place with a length-scale of 200 m. The model also includes an explicit ligand tracer for
complexing Fe, with ligand sources being from particulate organic carbon remineralization and dissolved organic
matter production. Ligand sinks are scavenging, uptake by phytoplankton, ultraviolet radiation, and bacterial
uptake or degradation.
This study is based on 5 sets of simulations for identifying contributions from different sources of DFe: control
simulation (CTRL); and simulations that individually remove DFe supply from atmospheric depositions (NATM),
sediments (NSED), rivers (NRIV) and hydrothermal vents (NVNT). Differences between CTRL and NATM
simulations indicate the biogeochemical impacts solely due to atmospheric deposition of DFe and is referred to
as ATM. Similarly, biogeochemical impacts solely from sedimentary, river and hydrothermal DFe sources are,
respectively, referred to as SED, RIV and VNT cases. Simulations have been conducted in hindcast mode for 60



years using forcing from the Coordinated Ocean-ice Reference Experiments version 2 (CORE-II) dataset for the
years 1948-2007 (Large & Year, 2009). The CORE-II data includes interannual variability and consists of 6-
hourly temperature, air density, specific humidity, 10 m wind-speeds, and sea-level pressure from National
Centers for Environmental Prediction/ National Center for Atmospheric Research (NCEP/NCAR) Reanalysis
(Kalnay et al., 1996). Daily shortwave and longwave radiation are taken from Goddard Institute for Space Studies-
International Satellite Cloud Climatology Project radiative flux profile data (GISS-ISCCP-FD) (Zhang et al.,
2004). Monthly precipitation is combined Global Precipitation Climatology Project (GPCP, Huffman et al., 1997)
and Climate Prediction Center Merged Analysis of Precipitation (CMAP, Xie & Arkin, 1997) data. Monthly
streamflow since 1948 is based on gauge data and CLM model has been used to calculate the freshwater fluxes
(Dai et al., 2009). The present study uses the last 10 years of simulations, given its focus on impacts of DFe
sources on biogeochemistry of the upper 100 m of the oceans at seasonal scale.
**2.2    Observation data**
Monthly climatology for ocean temperature, salinity and nutrients have been obtained from World Ocean Atlas
2018 (WOA18) at 1ºx1º spatial resolution (Garcia et al., 2019). Monthly surface chlorophyll concentrations have
been obtained from the European Space Agency Ocean Color Climate Change Initiative (OC-CCI) version 5 at 4
km spatial resolution for the period 2003-2020 (Satyendranath et al., 2019). OC-CCI merges ocean color
information from multiple sensors: Moderate Resolution Imaging Spectroradiometer (MODIS, 2002-present),
Sea-Viewing Wide Field-of-View Sensor (SeaWiFS, 1997-2010), MEdium Resolution Imaging Spectrometer
(MERIS, 2002-2012) and Visible Infrared Imaging Radiometer (VIIRS, 2012-present). The product is bias-
corrected and quality-controlled, yielding much lower data gaps compared to individual sensors. Monthly
climatology of mixed layer depth (MLD) gridded at 1ºx1º spatial resolution has been obtained from Argo profiles
based on a hybrid algorithm that calculates a suite of MLDs using several criteria, such as gradient/threshold
method, maxima or minima of a particular property, intersection with seasonal thermocline (Holte et al., 2017).
The resulting patterns are analysed to yield final MLD estimates. To explore ocean surface circulation, Ocean
Surface Current Analysis Real-time (OSCAR) data at 0.33ºx0.33º spatial resolution and 5-day temporal resolution
has been used. Horizontal velocities are measured using sea surface heights, ocean surface winds, and sea surface
temperatures, thereby accounting for flows due to geostrophic balance, Ekman dynamics, and thermal wind
(Dohan & Maximenko, 2010).

To examine the ability of CESM to realistically simulate the variation in DFe concentrations in the upper 100 m
over the northern IO, this study uses DFe profile compilations by Tagliabue et al. (2012) and the GEOTRACES
Intermediate Data Product 2021 (Schlitzer et al., 2021). To these, published data from Moffett et al. (2015) has
also been added, comprising DFe data collected in the AS during September 2007.  The DFe estimated in these
data are based on filtration of seawater through filter sizes between 0.2-0.45 µm.

**3    Results and discussions**
First, the performance of CESM-POP2 simulations with respect to observations over the northern IO is examined.
Next, the contributions of different DFe sources to upper ocean DFe concentrations, phytoplankton blooms and
patterns of nutrient limitations is discussed. Finally, the paper explores how different sources of DFe can influence
the total DFe budget across selected biophysical regimes over the northern IO.



### 3.1     Model evaluation

In this section CESM simulation (for CTRL case) of physical parameters as well as nitrate and chlorophyll concentrations over the upper 100 m of the northern IO is evaluated. Except for MLD, ocean currents, and chlorophyll, all modeled parameters have been compared with WOA18 observations. Simulated MLDs are compared with Argo-based values of Holte et al. (2017), ocean currents are compared with OSCAR data, and chlorophyll concentrations are compared with OC-CCI observations. In general, CESM shows good correspondence with observations of seasonal cycle of temperature, salinity and MLD. However, there is positive temperature and salinity bias over IO (Figs. S1 and S2 in the Supplement). This warm bias over IO differs from the previous version of CESM, which has a cold bias in this region (Danabasoglu et al., 2020). Figure 1 shows seasonal climatology in CESM simulations and observations, for MLD, nitrate concentrations, surface ocean currents, and chlorophyll concentrations. Overall, CESM simulates the main features of surface ocean circulation and spatio-temporal variations in MLD well. There are some deviations, such as a much stronger simulated Somali Current along the northeast coast of Africa, especially during the southwest monsoon season, which can lead to strong advection of upwelled nutrients away from this region. CESM also simulates a stronger South Equatorial Current during southwest monsoon, which occupies a broader region compared to observations and leads to a stronger westward flow in the model between 0-5ºS latitude. The net result of the warm and positive salinity bias is that CESM simulates much deeper MLD than observations throughout the year across the study domain. Averaged annually, the largest overestimation (of ~40 m) is over the equatorial IO particularly during the spring and fall intermonsoon months, when the Wyrtki Jet is prevalent over the region (Figs. S2 e-f). Additionally, MLD overestimation of ~45 m is also seen over the AS during February-March and the southern tropical IO during September-October, both associated with winter-convection.

With respect to the seasonal cycle of nitrate, CESM has the least bias over AS followed by BoB (Figs. 1a-d and S3), but its performance is comparatively lower over the equatorial IO and southern tropical IO. For example, WOA18 data shows the highest value of nitrate over southern tropical IO in January, whereas in CESM simulation the highest nitrate concentration is shifted to April-June associated with mixed layer deepening. On the other hand, CESM simulates a much weaker seasonal cycle of nitrate over the equatorial IO compared to WOA18 observations. These regions, over southern tropical IO and the equatorial IO, where CESM fares poorly also have fewer nutrient profile observations compared to AS and BoB. For example, no more than 10 nitrate observations are available in a grid-point over the southern tropical IO and equatorial IO, whereas there are several grid-points over the AS where more than 30 observations are available. Overall, CESM simulations underestimate nitrate with respect to WOA18 data for the upper 100 m of the water column.



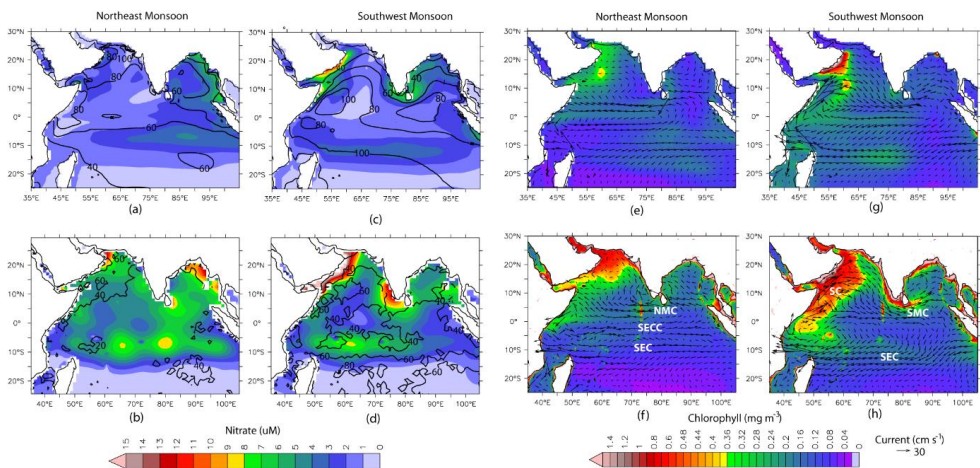

**Figure 1: Comparison of CESM-CTRL simulated variables (upper panels) with observations (lower panels) for northeast monsoon (a,b,e,f) and southwest monsoon (c,d,g,h). Shading in (a-d) are nitrate concentrations averaged for upper 100 m and the black contours are the mixed layer depth (m). Shading in (e-h) are surface chlorophyll concentrations and the vectors are the surface currents. SEC: South Equatorial current, SECC: South Equatorial Counter Current, NMC: Northeast Monsoon Current, SMC: Southwest Monsoon Current, SC: Somali Current.**

Turning to chlorophyll concentrations, CESM simulations capture the main characteristics of the seasonal cycle and its spatial distribution over the northern IO (Figs. 1e-h and S3), with certain biases and shifts in the timing of the peak blooms. For example, over the BoB, the model has difficulty in capturing the temporal evolution of chlorophyll concentrations. Over the AS and the equatorial IO, peak bloom in the simulations occurs in September, in contrast to July in the observations. Similarly, over the southern tropical IO, the peak bloom is delayed in the model to October as compared to its appearance in July in observations. Most of the AS and the BoB show underestimation (~ -60%) in simulated chlorophyll concentration with respect to OC-CCI values. Such underestimation of major nutrients and chlorophyll over most of the northern IO are common to many modelling studies where coastal regimes and mesoscale processes are not adequately captured without finer spatial resolution (e.g., Dutkiewicz et al., 2012; Ilyina et al., 2013; Long et al., 2021; Moore et al., 2013b; Pham & Ito, 2021). For example, a modelling study by Resplandy et al. (2011) has shown that eddy-induced vertical transport is responsible for ~40% of nitrate fluxes in the winter convection regions of the AS during the late northeast monsoon. The study also showed that mesoscale eddies can account for 65-91% of vertical and lateral advection of nitrate in the upwelling regions of the AS during the southwest monsoon. Additionally, the positive MLD bias simulated by CESM can trigger light limitation of phytoplankton growth, leading to underestimation of chlorophyll. If the threshold depth for photosynthesis is considered as the depth of the isolume given by 0.415 mol quanta $m^{-2}$ $day^{-1}$ ($Z_{0.145}$, Boss & Behrenfeld, 2010; Letelier et al., 2004), then the CESM simulated MLD is deeper than the $Z_{0.145}$, leading to light limitation of phytoplankton growth over the entire AS and large parts of BoB throughout the year (Fig. S4). During the southwest monsoon, almost the entire domain experiences light limitation, especially off the coast of Somalia and the southern tropical IO.

CESM simulations of DFe are evaluated next, using all available *in situ* DFe concentration data for upper 20 m of the ocean, for different seasons. In addition, distribution of DFe along selected transects for the upper 100 m



are studied: (1) CLIVAR cruise 109N along the eastern IO during April 2007; and (2) GEOTRACES cruises GI-
01, GI-02, GI-04 and GI-05. While CESM simulates the general pattern of DFe distribution over the northern IO
reasonably well, DFe variation with depth and with increasing distance from the coast is stronger in simulations
than in observations. For upper 20 m, correlation between observed and simulated DFe concentrations is 0.41
(Figs. 2a-d). The coefficients for correlation between observed and simulated DFe for GEOTRACES and
CLIVAR transects vary between 0.64 and 0.38 (Fig. 2e). All these correlation coefficients are significant at 95%
confidence level. This indicates that CESM is able to reproduce the north-to-south gradient in DFe concentrations,
the comparatively low DFe concentration west of 65ºE over the AS, as well as increases in DFe with depth over
both the eastern and western IO reasonably well.
Figures 2 f and g show two examples of variation of DFe distribution with latitude and depth along the eastern
and western IO, respectively. The model overestimates DFe values, especially to the north of the equator and at
depths greater than 50 m. Such overestimation of DFe over the northern IO in CESM could result from a variety
of factors, like source strength, assumed solubility of iron, biases in dissolved oxygen concentrations or ligand
concentrations, and uncertainties in the removal of DFe by biological uptake as well as scavenging. Specific
attribution for the overestimation of simulated DFe is beyond the scope of this paper. Dust deposition is one
possible factor leading to overestimation of simulated DFe. However, due to sparse dust deposition observations
available over this region, it is difficult to come to conclusion about its role in CESM-simulated DFe bias over
this region. Using Dust Indicators and Records of Terrestrial and MArine Palaeoenvironments (DIRTMAP)
version 2 database of modern day dust deposition (Kohfeld & Harrison, 2001) an attempt has been made here to
understand CESM bias in dust deposition over AS. Median dust deposition values from DIRTMAP ranges
between ~14 g m$^{-2}$yr$^{-1}$ over the western AS (40º-60ºE), ~7 g m$^{-2}$yr$^{-1}$ over the central AS (60º-70ºE) and ~20 g m$^{-2}$yr$^{-1}$ over the eastern AS (70º-80ºE) (Kohfeld & Harrison, 2001). Corresponding median values of dust deposition
over these locations from CESM model are 5 g m$^{-2}$yr$^{-1}$, 9 g m$^{-2}$yr$^{-1}$ and 14 g m$^{-2}$yr$^{-1}$ respectively. It is important to
note here that DIRTMAP represent dust depositions estimates for a specific location using a wide range of
methods, while CESM depositions are averaged over ~100 km. Over the eastern IO, using mixed layer dissolved
Al concentrations dust depositions have been estimated to be 0.2-3.0 g m$^{-2}$yr$^{-1}$ between 20ºS to 10ºN latitude
(Grand et al., 2015). In a separate study, based on Al concentrations in the aerosol, Srinivas and Sarin (2013) have
estimated dust dry-deposition flux of 0.3-3.0 g m$^{-2}$yr$^{-1}$ over BoB. Dust deposition from CESM is on the lower end
of this range varying from 1.1 g m$^{-2}$yr$^{-1}$ over the northern BoB to 0.2 g m$^{-2}$yr$^{-1}$ near the equator. Sediment traps
deployed at shallow depths over the BoB have recorded annual lithogenic fluxes varying from the northern to the
southern bay as ~15 g m$^{-2}$yr$^{-1}$ (~89.5ºE, 17.5ºN) to ~4 g m$^{-2}$yr$^{-1}$ (87ºE, 5ºN) (Unger et al., 2003). The corresponding
variations in CESM dust deposition are ~9 g m$^{-2}$yr$^{-1}$, to ~2 g m$^{-2}$yr$^{-1}$. Thus, overall, there is possibly some
underestimation of dust deposition over the northern IO, which might not explain positive DFe bias in CESM
simulations. Due to unavailability of measurements, it is very difficult to quantify the importance of other sources
of DFe in contributing to positive DFe bias in CESM simulations.



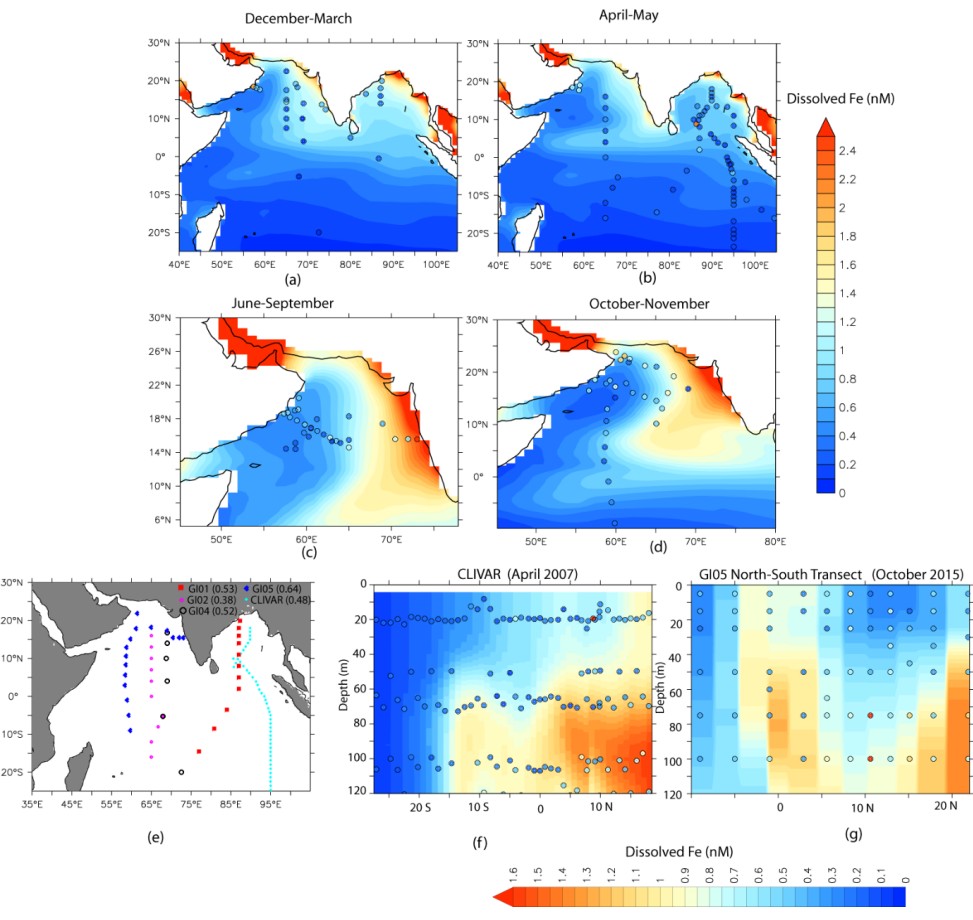

**Figure 2: Comparison of CESM-CTRL simulated DFe (shading) with the observations (filled circles) compiled from various cruises. The spatial distribution maps in (a-d) consider season-wise DFe distribution averaged over the upper 20 m. (e) The different cruise tracks from which DFe measurements have been used are marked. The numbers within the parentheses are the correlation coefficients between observed and simulated DFe for each cruise. The vertical transects in (f-g) show DFe gradients in the water column over (f) the eastern Indian Ocean and (g) the western Indian Ocean.**

It is seen that CESM consistently overestimates dissolved oxygen over the northern IO with respect to the WOA18 concentrations (Fig. S5). This implies that overestimation of sub-surface DFe concentrations in the model does not originate in the magnitude and the spatial extent of poorly oxygenated sub-surface waters. The impact of organic ligands in maintaining DFe stock by preventing scavenging losses can introduce yet another notable source of bias in simulated DFe. Only one study has measured ligand concentrations over the northern IO, during the spring intermonsoon of 1995 (Witter et al., 2000). At 100 m depth, observed ligand concentration ranges from 1.47 nM over the western AS to 4.94 nM over the eastern AS. The corresponding values from CESM simulations range from 1.55 nM in the western AS to 1.19 nM over the eastern AS. However, it is not possible to conclude about the impact of ligands on simulated DFe biases based on a single study. With respect to scavenging losses, it is quite possible that underestimation of productivity over the northern IO can lead to corresponding bias in




scavenging losses. This is because the base scavenging rate in CESM, apart from depending on dust fluxes, is also
a function of sinking fluxes of particulate organic matter, biogenic silica, and calcium carbonate. For example,
averaged over a year, there is ~60% underestimation in CESM of surface chlorophyll concentrations over the
northern IO, which would impact the sinking fluxes of biogenic matter. This can reduce scavenging losses,
especially, when there is a likely underestimation of dust deposition by CESM. Underestimation of phytoplankton
biomass over the northern IO can also lead to underestimation of phytoplankton uptake losses of DFe in the upper
100 m, which can be yet another source of overestimation of DFe.
To summarize, the ocean component of CESM model has deeper MLD than observations, underestimates nitrate
and chlorophyll and overestimates DFe concentrations. It is difficult to come to a definitive conclusion regarding
the importance of source strength in explaining the positive bias in DFe. It is quite possible that underestimation
of scavenging losses of excess DFe and biological uptake play vital roles in explaining positive DFe biases in this
region. Still, the model simulates spatial and temporal patterns of ocean physical features, as well as variations in
chlorophyll concentrations, nitrate, and DFe concentrations over the northern IO reasonably well. This gives
confidence in using the model to study the iron cycle over the region. Taking the above understanding of strengths
and shortcomings of the model into account, the importance of different DFe sources with respect to
biogeochemistry of the upper 100 m of the northern IO is explored next.

**3.2     Contribution of multiple iron sources**

Figure 3 summarizes the contributions of different sources to annually averaged DFe concentration. Source-wise
DFe contributions for northeast and southwest monsoons are shown in Figs. S6 and S7 respectively. Overall, the
relative contribution from different sources to DFe is roughly the same across different seasons, except for the
somewhat higher contribution of atmospheric DFe during southwest monsoon compared to northeast monsoon.
This is because the arid and semi-arid regions surrounding the northern IO experiences maximum dust activity
from late spring to early southwest monsoon months (e.g., Banerjee et al., 2019; Léon and Legrand, 2003). In the
annual average, atmospheric deposition is the most important source of DFe over the northern IO and contributes
well above 50% of the total DFe concentrations (ATM case in Fig. 3b). Furthermore, atmospheric deposition
contributes more than 70% of DFe supply over most of the AS, southern BoB, and the equatorial IO. The location
of the intertropical convergence zone during northeast monsoon (~10°S latitude) determines the southern limit of
the influence of atmospheric deposition because southwards of the intertropical convergence zone there is a rapid
reduction in DFe concentrations. Dust is the predominant contributor to the atmospheric deposition flux of iron.
Over the northern AS, dust is mostly transported from Iran, Pakistan, Afghanistan, and the Arabian Peninsula,
whereas over southern AS dust from north-eastern Africa also becomes important (Jin et al., 2018; Kumar et al.,
2020). Over northern and southern BoB, the major sources of dust are the Indo-Gangetic Plain and northeast
Africa, respectively (Banerjee et al., 2019). Eastwards of 90°E, black carbon contributes ~50% to atmospheric
DFe flux during the northeast monsoon (not shown). The source of black carbon in this region is biomass burning
and fossil fuel combustion transported from the Indo-Gangetic Plain and Southeast Asia (Gustafsson et al., 2009;
Moorthy & Babu, 2006).





The second largest source of DFe is from continental shelf sediments (Fig. 3c), which become dominant in the
vicinity of the shelves. High sedimentary sources of DFe are characteristic of the Andaman Sea where incoming
rivers can contribute ~600 x $10^6$ T yr$^{-1}$ of sediments (Robinson et al., 2007). It has been estimated that terrestrial
sources contribute more than 80% to total organic carbon in the inner shelf region of the Gulf of Martaban,
adjacent to the Andaman Sea (Ramaswamy et al., 2008). Elsewhere, sedimentary contributions of ~20% to overall
DFe are found in CESM runs along the northern part of west coast of India and the eastern BoB. Within Ganga-
Brahmaputra system, which is responsible for discharge of ~11 x $10^8$ T yr$^{-1}$ of sediments, only 10% of sediments
is estimated to be transported longshore, with most of the sediments accumulating within the shelf and
subterranean canyon (Liu et al., 2009). Over the open ocean, sedimentary sources are most important within 10°-
15°S latitude where the South Equatorial Current is responsible for ~50% of DFe supply via advection from the
Indonesian shelf. During southwest monsoon, sedimentary contribution by the South Equatorial Current extends
farther westward (~70°E longitude, Fig. S7c) compared to the northeast monsoon (~80°E longitude, Fig. S6c).
Signatures of elevated Al due to sedimentary contribution is seen in ship-borne measurements (Grand et al., 2015;
Singh et al., 2020). In fact, such measurements have shown that the South Equatorial Current separates DFe-rich
oxygen-poor water of the northern IO from the DFe-poor oxygen-rich water of the southern tropical IO (Grand et
al., 2015).
River sources contribute negligibly to total DFe concentrations (Fig. 3d), except in the immediate vicinity of the
mouths of large river systems in the northeast BoB: the Ganges-Brahmaputra and the Irrawady-Sittang-Salween.
This is possibly because flocculation at the river mouth can quickly lead to near-complete losses of DFe compared
to other metals (Flegal et al., 1991; Sholkovitz, 1978). Hydrothermal vents also contribute negligibly to DFe
concentrations in the upper 100 m (Fig. 3e). The hydrothermal vents supplying DFe (often excess of 1.5 nM) in
the northern IO are located in the Central Indian Ridge and the Carlsberg Ridge (Chinni & Singh, 2022; Nishioka
et al., 2013; Vu & Sohrin, 2013), and largely influence DFe concentrations below 1000 m depths. The shallowest
hydrothermal plumes enriched with Fe are located between ~650-900 m in the Gulf of Aden (Gamo et al., 2015),
overlapping with the depth range at which the Red Sea watermass spreads along the western IO (Beal et al., 2000).
Since this watermass occupies progressively deeper depths with distance, sliding underneath Persian Gulf waters,
surface DFe values are not impacted by these shallower vents. This is in concordance with simulations of
Tagliabue et al. (2010) where, following 500 years of model integration, hydrothermal vents increase globally
averaged DFe concentrations by only ~3% in the depth range of 0-100 m.
The average contribution of different sources of iron to the upper 100 m is summarized for different open ocean
regions over the northern IO in Fig. 3f. Annually averaged atmospheric deposition is clearly the most important
source of DFe throughout the northern IO. This source accounts for almost the entire supply of DFe over the
equatorial IO. The exception to the dominant role of atmospheric deposition is the southern tropical IO, where
sedimentary sources of iron contribute ~40% to the upper ocean iron budget. Overall, river contribution is
generally ~1%, with slightly higher contributions in BoB and the southern tropical IO. Hydrothermal vents make
negligible contributions throughout the northern IO. Adding these four sources of DFe estimated from CESM
experiments does not yield the full 100% of the DFe source, possibly owing to non-linear effects associated with
iron removal processes as well as complexation by organic ligands.

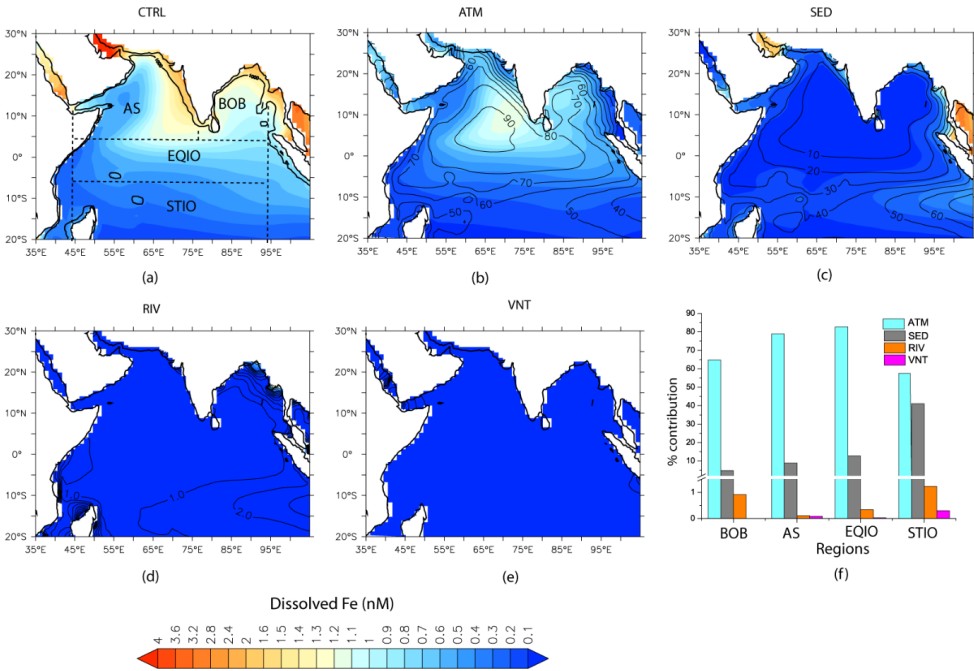

**Figure 3: Contribution of different sources of DFe averaged over the year to the total DFe concentrations over the upper 100 m. Shading in (a) shows total DFe concentration with all sources included and shadings in (b-e) shows DFe concentrations arising from individual source. Contours in (b-e) show the percentage contribution of each source to total DFe concentrations. (f) Bar chart depicting source-specific DFe contribution (in %) over Bay of Bengal (BOB), Arabian Sea (AS), equatorial IO (EQIO), and the southern tropical IO (STIO). These regions are marked by the dashed boxes in (a). The thick black contour in (a) traces the 1000 m bathymetry.**

### 3.3 Phytoplankton responses to multiple iron sources

In this section, the impact of different sources of DFe on phytoplankton growth is examined. Since river and hydrothermal sources make negligible contributions to the upper ocean iron concentrations, as shown above, these are not considered further.

### 3.3.1 Responses to atmospheric depositions

During the northeast and southwest monsoons, atmospheric DFe brings about increases in column-integrated chlorophyll concentrations over most of the northern IO (Figs. 4 a and c). The largest column-integrated positive response is seen in the western AS (west of ~65°E longitude) throughout the year, where atmospheric DFe accounts for more than ~20% of the column-integrated chlorophyll concentration and more than 50% of surface chlorophyll concentration (Fig. S8). This region comes under the influence of upwelling during the southwest monsoon and mixed layer deepening due to winter convection during the northeast monsoon, which can supply macronutrients required for phytoplankton growths (Madhupratap et al., 1996; Morrison et al., 1998). The other region displaying a strong positive response is the southern tropical IO during June-September, where atmospheric DFe contributes ~20% (~35%) of the column (surface) chlorophyll concentration. This is the time of the year when deep mixed layer leads to entrainment of nutrients into the surface layers (Kŏne et al., 2009; Lévy et al.,





2007). In contrast, there are some regions, like the northern and western AS, the west coast of India and large
parts of the BoB and the eastern IO, which in spite of receiving high atmospheric DFe hardly experience any
chlorophyll response. These regions show <1% increase in column chlorophyll concentrations and generally
coincide with high sedimentary iron input. This is discussed further in Section 3.3.3.

Species-wise decomposition shows that the increases in chlorophyll during both northeast and southwest
monsoons are driven by increases in diatoms and declines in small phytoplankton (Fig. 5). For example, over the
western AS and southern tropical IO, diatoms increase by at least 40% and small phytoplankton populations
decline by at least 50%. Diatoms outperforming other phytoplankton species has been previously witnessed in *in*
*situ* iron fertilization experiments (de Baar et al., 2005). This is due to the large cell size of diatoms enabling
higher cellular uptake of iron and also the ability of diatoms for luxury iron uptake, which enables them to
outcompete other species in a bloom (Sunda & Huntsman, 1995). An exception is the equatorial IO, where the
positive response of chlorophyll arises from growth of small phytoplankton. In general, this region has very low
levels of macronutrients and is dominated by picoplankton (Vidya et al., 2013). Those regions exhibiting <1%
increase in phytoplankton in response to atmospheric DFe, in contrast, are characterized by proliferation of small
phytoplankton and reductions of diatoms. Although diazotrophs show positive response to atmospheric DFe
addition throughout the region, this group constitutes only ~1% of total phytoplankton biomass. Such shifts in
phytoplankton community structure in response to DFe additions are also corroborated by *in situ* experiments
over the northern IO. For example, a nutrient addition experiment over the northern AS during northeast monsoon
period has shown that the maximum positive phytoplankton response takes place due to nitrate+DFe addition
(instead of only DFe addition), accompanied by around four-fold increases in coccolithophores, pennate and large
centric diatoms (Takeda et al., 1995). Ship-board iron addition experiments over the AS during the southwest
monsoon resulted in proliferation of visible colonies of haptophyte *Phaeocystis sp.* due to silicate-limitation
(Moffett et al., 2015). Over the eastern IO, where both macronutrients and micronutrients are low, nutrient spiking
with nitrogen, phosphorus, and iron resulted in increase of Prochlorococcus, Synechoccus, as well as Eukaryotes
(Twining et al., 2019).
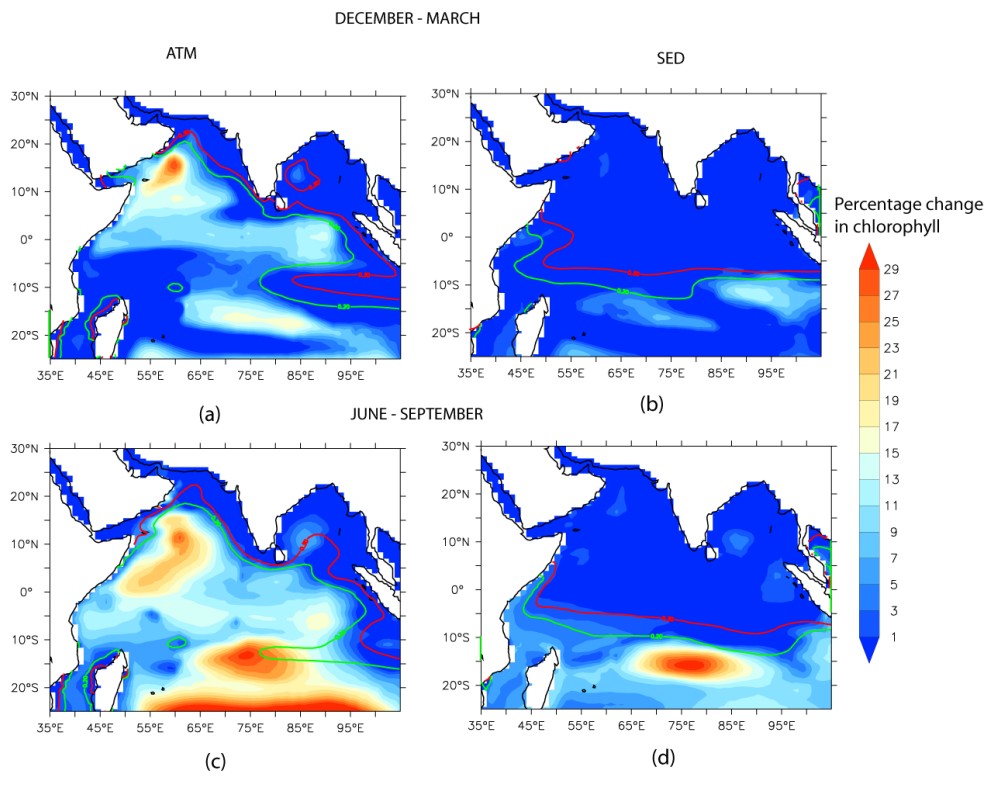


**Figure 4: Percentage contribution of (a and c) atmospheric and (b and d) sedimentary sources of iron during (a and b)**
**the northeast monsoon and (c and d) the southwest monsoon to column-integrated (0-100 m depth) chlorophyll**
**concentrations. Green and red contours show background DFe concentrations of 0.2 nM and 0.3 nM respectively. For**
**the ATM (SED) case, background DFe is obtained from NATM (NSED) simulation.**




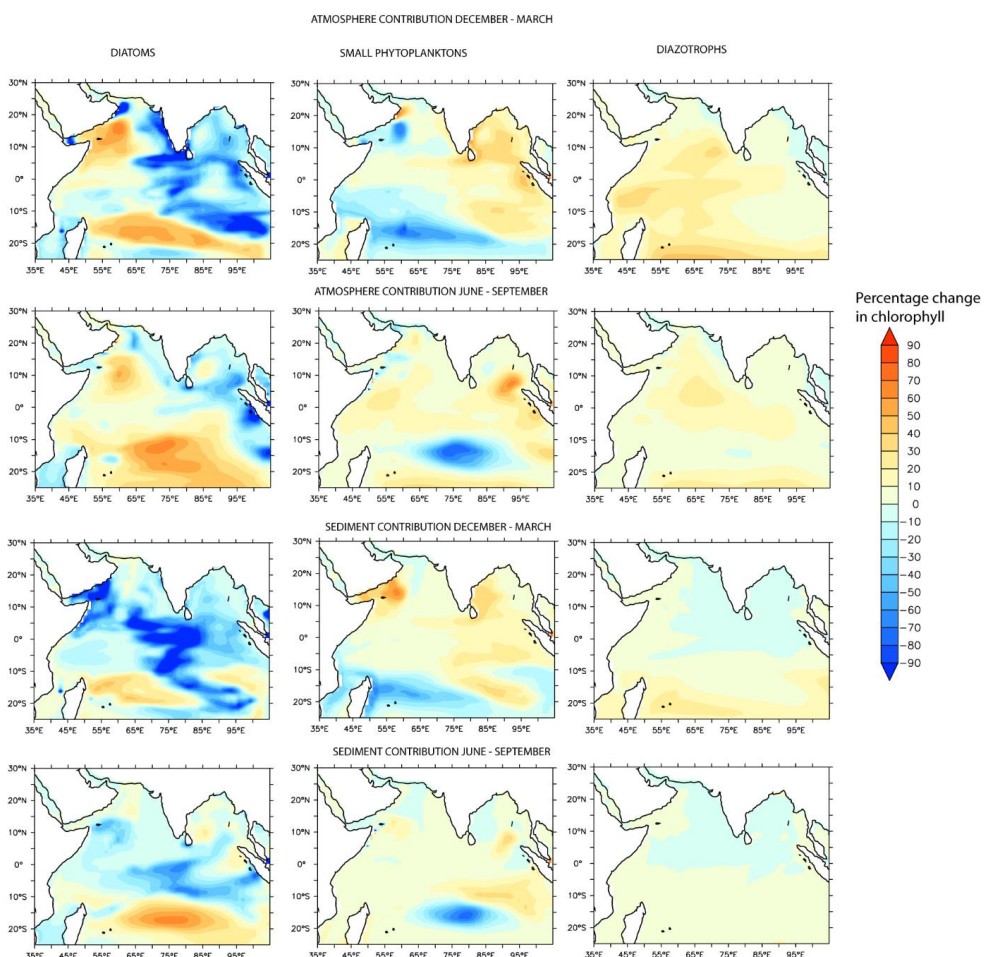

**Figure 5: Species-wise percentage contribution to column chlorophyll (0-100 m) response associated with atmospheric and sedimentary sources of DFe.**

### 3.3.2 Responses to sedimentary sources of iron

As shown in Fig. 3, sedimentary sources supply less than ~20% of DFe north of ~10ºS latitude, whereas between 10º-15ºS latitude sedimentary iron can contribute to almost half the total DFe concentrations. Unlike atmospheric sources, sedimentary supply of DFe is mostly confined to regions adjoining continental shelves and islands from where they are introduced to the open ocean by seasonally varying currents. In general, sedimentary sources make modest contribution to column productivity (<1% of chlorophyll anomalies) to the north of ~10ºS latitude as described above. This is because high dust deposition to the north of the intertropical convergence zone results in high background DFe concentrations and controls productivity (see also Section 3.3.3). Sedimentary sources trigger the strongest positive phytoplankton response over the southern tropical IO region during June-September, where sedimentary DFe advected by the South Equatorial Current can facilitate more than 20% increase of the upper 100 m chlorophyll concentrations and ~40% increase at the surface. As noted in Section 3.2, although





atmospheric deposition contributes nearly half of the total DFe addition to this region, the total iron deposition
here is low (<0.2 nM). The phytoplankton response over the southern tropical IO is dominated by an increase in
diatoms, which contribute to more than 60% of total phytoplankton biomass (Fig. 5). In contrast, over the regions
experiencing <1% chlorophyll increase, there is a shift from diatoms towards small phytoplankton species (Fig.
5). For example, there is more than 80% reduction in diatoms and 50% increase in small phytoplankton over the
western AS. Other current systems such as the poleward flowing Somali current, the eastward flowing Southwest
Monsoon Current and its southward extension along the west coast of Indonesia also transport sedimentary DFe
to the open ocean, but such advection supports only ~5% phytoplankton biomass.
### 3.3.3   Role of background nutrients in phytoplankton responses to external iron

It emerges from the previous sections that there is heterogeneity in the phytoplankton response to atmospheric
and sedimentary sources of DFe. The regions of highest DFe input from a specific source are not always the
regions where strongest phytoplankton responses are evoked. What explains these differing patterns of
phytoplankton response? To examine this, patterns of nutrient limitations and iron supply from an external source
with respect to background DFe and nitrate ($NO_3$) concentrations are examined. In considering the phytoplankton
response to atmospheric sources (ATM case), background DFe is taken from the simulation without any
atmospheric source (NATM). Since river and hydrothermal sources make negligible contributions to DFe over
this domain, high levels of DFe in NATM mainly arise in regions where sedimentary sources are important.
Similarly, for estimating phytoplankton response to sedimentary sources (SED case), background DFe is taken
from simulation without any sedimentary source (NSED).
Generally, those regions experiencing greater than 1% increase in chlorophyll in response to atmospheric
(sedimentary) sources coincide with background DFe concentration <0.2-0.3 nM and high background $NO_3$:DFe
ratio from the NATM (NSED) simulation. For example, in NATM simulation, iron serves as the dominant nutrient
that limits productivity over the entire northern IO, with diatoms experiencing stronger iron limitation compared
to other phytoplankton groups (Fig. S9). Iron limitation is particularly severe over central and southern AS,
equatorial IO and the southern tropical IO. In NSED case, there is a switch from nitrate limitation to the north of
the intertropical convergence zone to iron limitation to the south of the intertropical convergence zone (Fig. S10).
While iron stress is alleviated with addition of external DFe, there is a shift towards macronutrient, especially
nitrate, limitation (Fig. 6). South of ~15°S latitude continues to experience iron limitation during June-September
due to very low dust deposition. In contrast, regions where chlorophyll increase is <1% following DFe addition
are characterized by nitrate limitation in NATM/NSED simulations and external DFe cannot alleviate this primary
nutrient limitation.  This is further illustrated in Fig. 7 where $NO_3$:DFe ratio is plotted against background DFe
concentrations. Positive chlorophyll response is elicited in regions of lowest background DFe and highest
$NO_3$:DFe ratio. Over the world oceans, a wide range of DFe:C ratio has been observed for diatoms, ranging from
$4.3 \times 10^{-5}$ for DFe-replete conditions to $2.0 \times 10^{-6}$ for DFe-deplete conditions (de Baar et al., 2008). Assuming
C:N ratio of 117:16 (Anderson and Sarmiento, 1994), range of N:DFe ratios obtained are ~3000 and ~68000,
respectively, for DFe-replete and DFe-deplete conditions. Similarly, by considering iron limitation taking place
for DFe:C ratio of $1 \times 10^{-5}$ for open ocean species based on laboratory experiments (Sunda & Huntsman, 1995)
and C:N ratio of 106:16, Measures and Vink (1999) have estimated that iron limitation over the AS takes place at
$NO_3$:DFe ratio greater than ~15000. In CESM simulations >1% increase in chlorophyll takes place when initial





NO$_3$:DFe ratio is more than 10,000 corresponding to Fe-limitation scenario (Fig. 7). With the addition of DFe
from atmospheric or sedimentary sources, the NO$_3$:DFe ratio reduces to even less than ~4000 in some cases,
thereby leading to N-limitation. Previously, iron addition experiments in AS during the southwest monsoon have
shown that the positive chlorophyll response depends on initial nitrate concentrations, with this response
increasing in magnitude with higher initial nitrate concentrations (Moffett et al., 2015). In summary, the initial
NO$_3$:DFe ratio sets the ultimate limit to the magnitude and distribution of phytoplankton response following
external DFe additions.

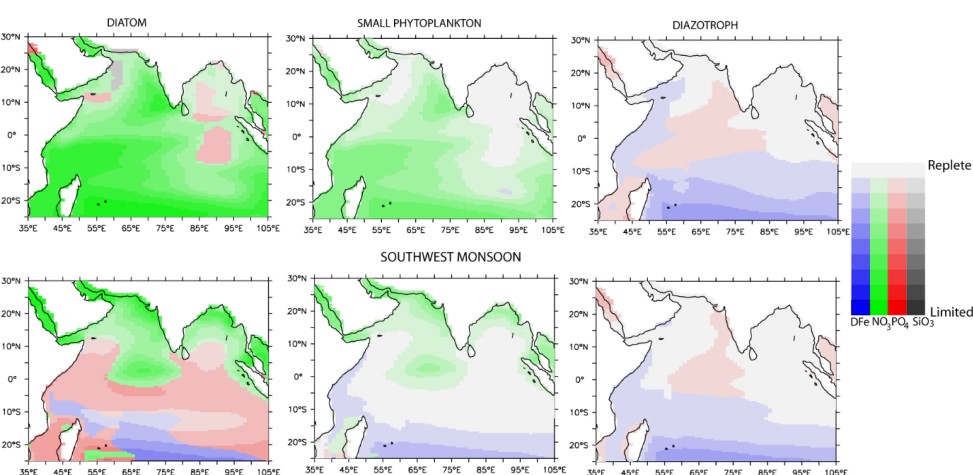



**Figure 6: Patterns of surface nutrient limitations for different phytoplankton functional types from CTRL simulation.**
**Green: nitrate; blue: iron; red: phosphate; grey: silicate limitations.**

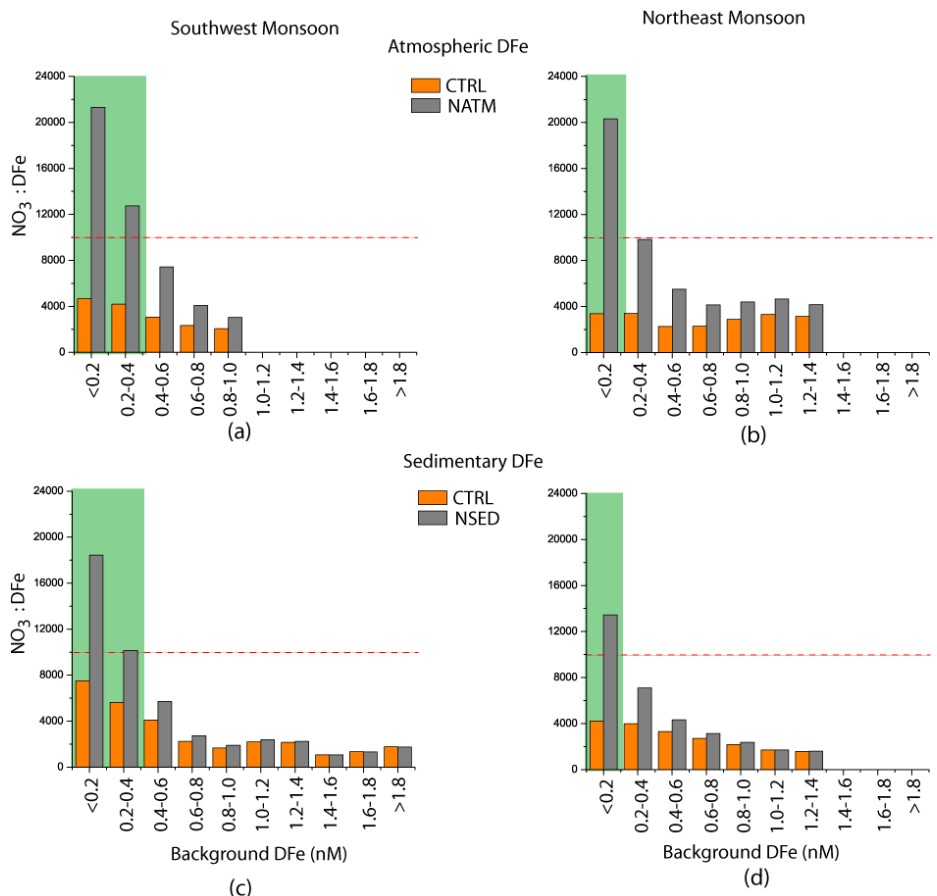


**Figure 7: Relation between background nutrients and phytoplankton response for atmospheric (a and b) and sedimentary (c and d) sources of DFe during (a and c) southwest monsoon and (b and d) northeast monsoon. The horizontal axis shows background DFe concentrations. The orange columns show NO$_3$:DFe ratio for CTRL case and grey columns show NO$_3$:DFe ratio for (a-b) NATM and (c-d) NSED cases. The red dashed lines show the location where NO$_3$:DFe ratio is 10,000: below this value N-limitation prevails in CESM. Green shades highlight the regions where >1% increase in chlorophyll following DFe addition from a specific source is induced.**

To sum up, atmospheric deposition is the most important source of DFe to the upper 100 m over the entire northern IO, followed by sedimentary sources. While atmospheric DFe is deposited over wide areas of the open ocean, sedimentary DFe fluxes arise only from continental shelves and are transported to open oceans through advection by currents. River and hydrothermal sources make negligible contributions to the total iron budget in the upper 100 m. The primary response to atmospheric DFe is an increase in column-integrated phytoplankton biomass over most of the northern IO. In contrast, sedimentary source of iron is responsible for increases in column-integrated phytoplankton biomass mainly to the south of the intertropical convergence zone, where dust depositions are low. In general, significant positive responses of phytoplankton to addition of DFe are simulated only where low levels of background DFe concentrations and high values of background NO$_3$:DFe ratio are present. Otherwise, nitrate becomes the limiting nutrient once DFe is added. The simulations also show that positive chlorophyll response

to addition of DFe generally involves proliferation of diatoms, except over the equatorial IO where small
phytoplankton increase is seen.

### 3.4    Iron budgets across different bio-physical regimes

This section explores the main processes controlling DFe budget with respect to the role of atmospheric and
sedimentary sources over different bio-physical regimes of the northern IO: (1) the western AS, (2) the southern
BoB, (3) the central equatorial IO and (4) the central southern tropical IO. These regions encompass a wide range
of productivity, with the first region being highly productive with OC-CCI chlorophyll exceeding 1.5 mg m$^{-3}$. The
southern BoB and central southern tropical IO are moderately productive. Lastly, the central equatorial IO is
oligotrophic with surface chlorophyll concentration being ~ 0.1 mg m$^{-3}$. The locations of these regions along with
CESM simulated seasonal cycles of mixed layer depths, chlorophyll and dust depositions are shown in Fig. 8.

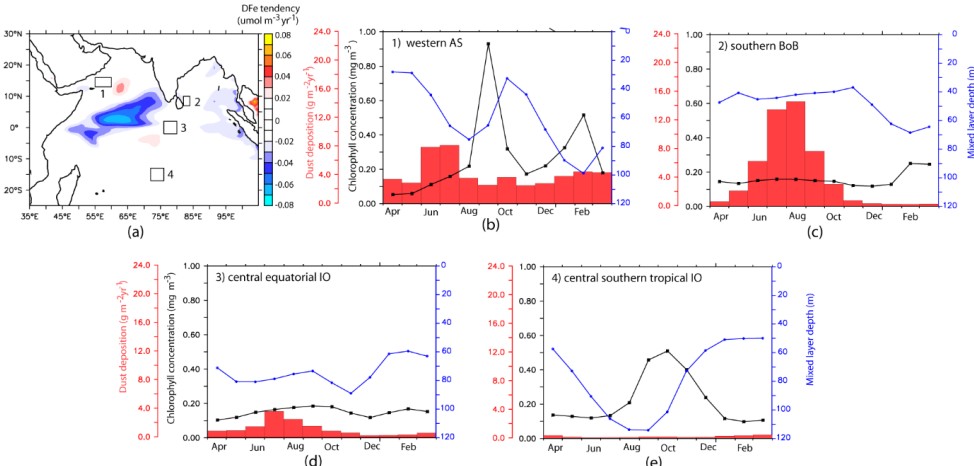


**Figure 8: (a) Net DFe tendency averaged over the upper 100 m for the study period. The boxes indicate the regions**
**chosen for further studying DFe budget in Section 3.4. (b-e) Seasonal cycle of dust deposition (red columns), mixed**
**layer depth (blue curves) and chlorophyll concentrations (black curves) from CESM-CTRL case for the four regions**
**marked in (a).**

The net dissolved iron tendency (TEND$_{DFe}$) is calculated as:

$$\text{TEND}_{\text{DFe}} = \text{EXT} + \text{ADV} + \text{MIX} + \text{BIO} \qquad (1)$$

where the source terms on the right describe dust/sediments/rivers/vents (EXT), horizontal and vertical advection
(ADV), horizontal and vertical mixing (MIX) and biological sources/sinks (BIO). Advection includes explicitly
resolved velocity as well as an additional "bolus" velocity from parameterization of mesoscale eddies (Gent &
McWilliams, 1990). Vertical mixing includes a tracer gradient dependent term for cross-isopycnal mixing and a
non-local mixing term, which accounts for mixing due to convective and shear instabilities (Large et al., 1994).
Lateral mixing involves parameterization of mesoscale eddy-induced horizontal diffusion along isopycnal
surfaces (Redi, 1982). The BIO term includes DFe losses due to biological iron uptake and scavenging, recycling





of iron back to the pool via remineralization, and iron released from phytoplankton and zooplankton losses and
grazing.
**3.4.1    Western Arabian Sea**
The western AS, off Oman and Yemen coastlines (considered here as 13º-16ºN and 55º-60ºE), is the most
productive region in the northern IO. Primary productivity in the western AS is highest during southwest monsoon
(Fig. 8b), during which alongshore southwesterly winds lead to upwelling and bring subsurface nutrients from
depths of ~150-200 m (Morrison et al., 1998). Some of this upwelled water advects eastwards, transporting
nutrients that enhance productivity in the central AS (Prasanna Kumar et al., 2001). The region also experiences
a secondary bloom during northeast monsoon due to winter convection that deepens the mixed layer. Integrated
over depths of the euphotic zone, average primary productivity over the western AS during mid and late southwest
monsoon is estimated at $135\pm10$ mmol C m$^{-2}$ d$^{-1}$ and $110\pm11$ mmol C m$^{-2}$ d$^{-1}$ respectively (Barber et al., 2001). In
comparison, primary productivity over the western AS during mid and late northeast monsoon is $137\pm13$ mmol
C m$^{-2}$ d$^{-1}$ and $88\pm4$ mmol C m$^{-2}$ d$^{-1}$ (Barber et al., 2001).  Although this region encounters high dust deposition
(Haake et al., 1993; Mahowald et al., 2009), *in situ* measurements have hypothesized possible iron limitation
during late southwest monsoon because upwelled water is drawn from above the iron-rich sub-oxic zone (Naqvi
et al., 2010).
The largest peak in dust deposition is during southwest monsoon, followed by a second peak during northeast
monsoon (Fig. 8b). Accordingly, the upper ocean DFe concentration is highest during southwest monsoon and is
dominated by atmospheric sources (Fig. 9). Sedimentary contribution, although much lower, peaks during late
southwest monsoon and fall intermonsoon months. Throughout the year DFe concentration increases with depth,
thus pointing to consumption by phytoplankton at the surface. Vertical advection and vertical mixing are the most
important physical mechanisms governing DFe supply within this region during southwest monsoon (Fig. 9).
These processes begin to strengthen from May onwards to reach their peak during June-July and decrease
thereafter. Decomposing DFe advection tendency into tendencies arising from gradients in tracer distribution
(DFe´) and velocity convergence (U´) respectively, it is seen that vertical advection of DFe arises from DFe´ and
U´ in equal magnitude. However, the former process is dominant in June and the latter process dominates during
July (Fig. S11). The maximum vertical advection of DFe is centered around 80 m depth and progressively reduces
at shallower depths, as the vertical velocity reduces towards the surface. Vertical mixing prevailing in the upper
40 m brings this vertically advected DFe from subsurface to the surface. Furthermore, horizontal advection plays
an important role in redistributing this DFe supplied by vertical processes, with contributions from horizontal U´
being at least twice as large as DFe´. During spring and early southwest monsoon, northeastward horizontal
advection removes atmospheric deposited DFe throughout the upper 100 m, while aiding the supply of
sedimentary DFe from Somalia and Omani continental shelves to the western AS. Later in the year as the
southwest monsoon current circulation is established, and meridional currents along the western AS become
stronger, its effect is first evident in the south along the Somali coast and progresses northward with time. The
result is convergence of both atmospheric and sedimentary DFe in the western AS during July-September. During
northeast monsoon, vertical mixing driven by winter convection, with the mixed layer deepening to 100 m, is the
most important means of DFe supply, from both atmospheric and sedimentary sources, into the surface layer.



Additionally, horizontal advection by westward currents transports DFe from atmospheric deposition in the central
AS into the western AS.
Removal of DFe from the water column is mainly through biological uptake in the upper 40 m. Uptake of DFe by
small phytoplankton dominate biological uptake throughout the year, except during September-October when
diatoms uptake of DFe becomes significant (not shown). This signature of diatoms is also observed in opal fluxes
measured by sedimentary traps deployed near the western AS and has been attributed to lowering of zooplankton
grazing pressures during late southwest monsoon (Smith, 2001) as well as to silicate limitation of diatoms in
initially upwelled waters (Haake et al., 1993). In the subsurface layer, remineralization of sinking fluxes of
particulate iron peaking at ~50 m replenishes the DFe pool during the latter part of the productive months (Fig.
S15a). Iron so released is made available to the surface layer via mixing or advection, thereby playing an important
role in maintaining surface DFe pool. Some of the remineralized DFe is further removed by scavenging, which
peaks at ~80 m during the productive months due to large fluxes of sinking particulate organic carbon, biogenic
silica, calcium carbonate and dust (Fig. S15a). Atmospheric deposition dominates biological source/sink of DFe
throughout the year, while sedimentary DFe is more important for biology during northeast monsoon months.



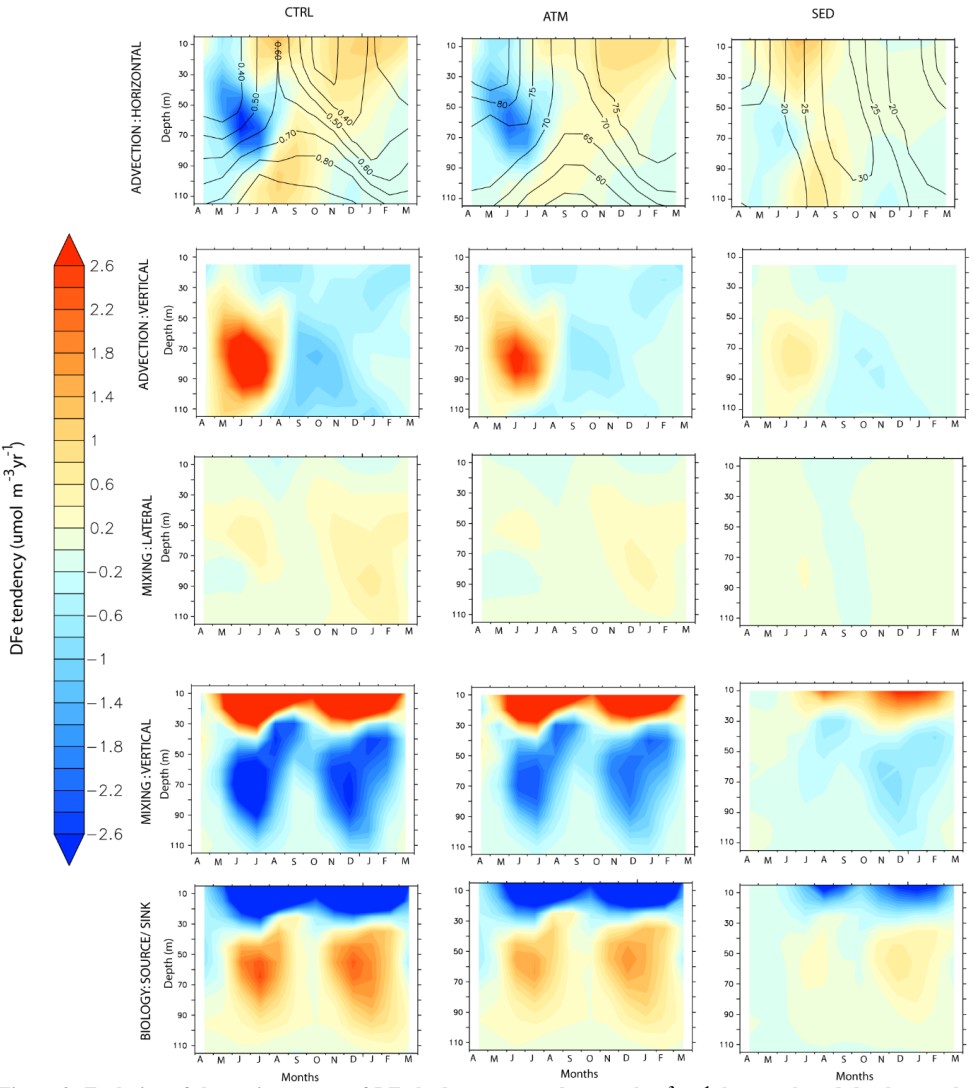

**Figure 9: Evolution of the various terms of DFe budget, expressed as μmol m⁻³ yr⁻¹, by month and depth over the**
**western Arabian Sea. Left panels: CTRL, Middle panels: ATM and, Right panels: SED case. The contours in the upper**
**panel for CTRL show evolution of DFe concentrations (nM), while the contours in the upper panels for ATM and SED**
**cases show the percentage contribution of each of these cases to total DFe concentrations in CTRL case.**

**3.4.2 Southern Bay of Bengal**
The region corresponding to the southern BoB (7º-10ºN and 82º-84ºE) is located to the east of Sri Lanka.
Compared to the rest of the BoB, freshwater flux from South Asian rivers reduces markedly in this region due to
advection of high salinity water from AS by the eastward flowing Southwest Monsoon Current (see Fig. 1h) as
well as upward pumping of saltier water by thermocline doming during the southwest monsoon season
(Vinayachandran et al., 2013). This leads to stronger biophysical coupling in the southern BoB, compared to the
rest of the bay, through erosion of the upper stable layer of freshwater capping. During southwest monsoon, the





Southwest Monsoon Current advects nutrients and chlorophyll from the upwelling regions along the southern tip of India and Sri Lanka into the southern BoB (Vinayachandran et al., 2004). Over the open southern BoB, to the east of Sri Lanka, cyclonic wind stress curl drives open ocean upwelling leading to shoaling of the thermocline that forms the Sri Lankan dome. This results in surface chlorophyll concentration between 0.3-0.7 mg m$^{-3}$ and strong subsurface chlorophyll maxima between 20-50 m where chlorophyll concentration can exceed 1 mg m$^{-3}$ (Thushara et al., 2019). A much lower magnitude of surface chlorophyll concentration (~0.18 mg m$^{-3}$, Fig. 8c) and subsurface chlorophyll maxima (~0.2 mg m$^{-3}$) at 40-60 m depth is simulated by CESM. During the northeast monsoon, CESM simulates a second bloom over this region associated with winter cooling and mixed layer deepening to ~60 m (Fig. 8c). This bloom has slightly higher magnitude, peaking at ~0.25 mg m$^{-3}$, compared to the southwest monsoon bloom. Surface chlorophyll data from OC-CCI also reveals the presence of northeast monsoon blooms (peak at ~0.25 mg m$^{-3}$), which during some years are of higher magnitude than southwest monsoon blooms. Argo data in this region also show signatures of mixed layer deepening during winter (not shown).

Overall, the highest DFe over this region is encountered during the late southwest monsoon and is dominated by atmospheric deposition (Fig. 10). Vertical advection is the most important process supplying DFe to the surface layers during spring and southwest monsoon months (Fig. 10). This is aided by a positive wind stress curl established over the region from March onwards. While vertical velocity is positive during the southwest monsoon over the entire depth considered, DFe supply by vertical advection is positive only for depths less than 50 m (Fig. S12). This is because the magnitude of upward velocity gradually reduces with depth, resulting in positive values of U´ upwards from 40 m depths. (Fig. S12). With the arrival of westward propagating Rossby waves to the western boundary of the BoB during October, upwelling favorable vertical motion collapses (Webber et al., 2018).

With respect to horizontal advection, it is seen that the magnitude and sign of convergence by the meridional component of the current mainly controls DFe supply over the southern BoB. This arises from the southward flowing current to the western flank of the Sri Lankan dome that supplies atmospheric DFe to this region. This DFe supplied by the southwards current, as well as DFe derived from upwelling, is removed by the energetic eastward currents during late spring to early fall intermonsoon months. During the rest of the year, the westward flowing currents supplies some sedimentary DFe from the Andaman Sea to the southern BoB. However, the much larger magnitude of dust deposition in the north-western BoB leads to overall negative tracer gradients and, thus, dilution of DFe by horizontal advection. The most important DFe supply mechanism during northeast monsoon is enhanced vertical mixing in the upper 20 m associated with deepening of mixed layer. Additionally, downwelling due to weakly negative wind stress curl during this time of the year removes DFe from the surface and favors its accumulation in the subsurface ocean. Lateral mixing complements DFe supply to the upper 20 m during fall and early northeast monsoon, especially from sedimentary sources.

Biological uptake removes DFe throughout the year from the upper 40 m especially during the southwest and northeast monsoon blooms (Fig. 10). DFe uptake in the upper 40 m is dominated by small phytoplankton during most of the year, except during northeast monsoon (not shown). Diatom DFe uptake, on the other hand, dominates the deep chlorophyll maxima present between 40-70 m throughout the year as well as within the surface layer during northeast monsoon months. Several studies have pointed to substantial nutrient uptake by diatoms in the central, coastal, and northern BoB due to riverine supply of silicates (Madhu et al., 2006; Madhupratap et al.,





2003). Remineralization of DFe as well as DFe release from grazing and mortality of phytoplankton and
zooplankton have a primary peak between 50 m-80 m during July-August and secondary peak during February-
March. On the contrary, scavenging removes DFe, with its effect peaking during July-August during blooms (Fig.
S15b).

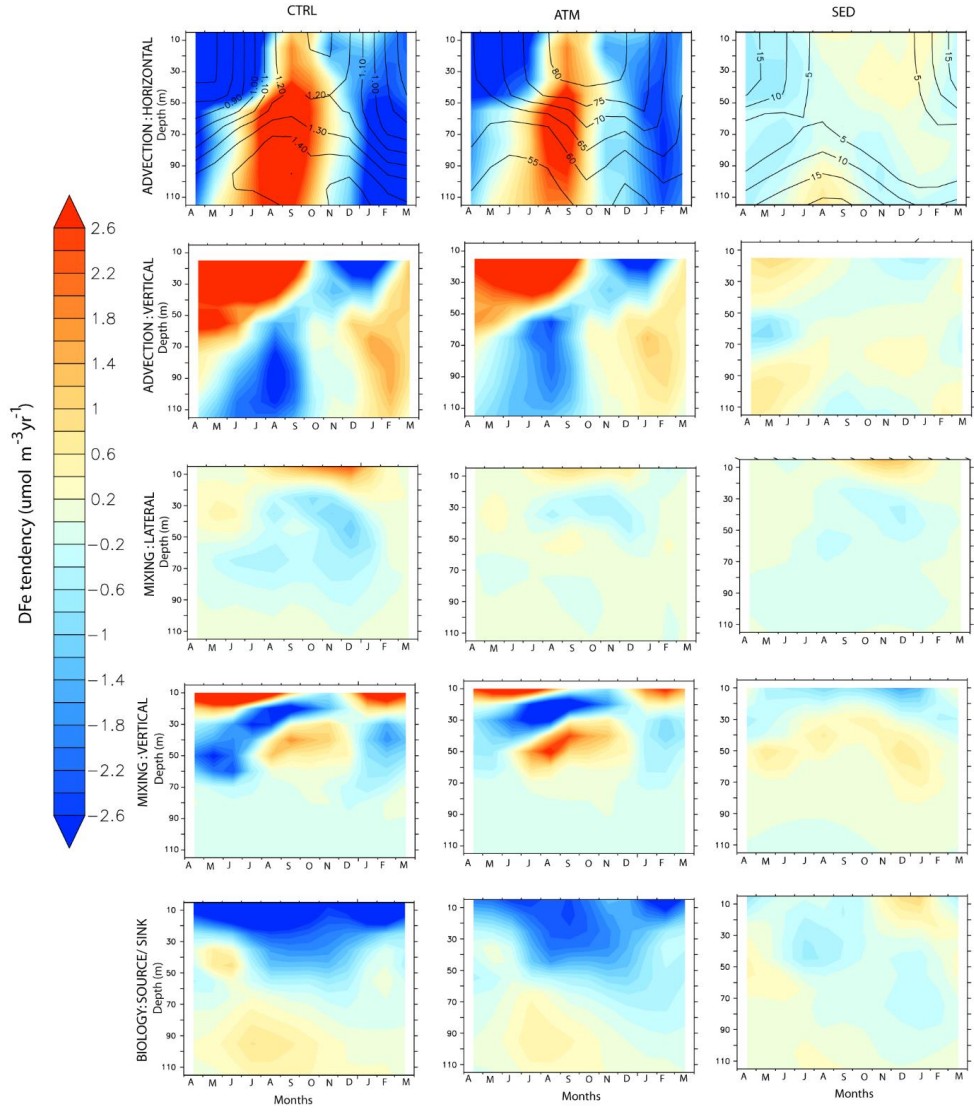

**Figure 10: Same as Figure 9, except over the southern Bay of Bengal.**

**3.4.3    Central Equatorial IO**
With chlorophyll concentrations around 0.1 mg m$^{-3}$ for most part of the year, the central equatorial IO (2ºS-2ºN
and 76º-80ºE) is the least productive of all the regions considered (Fig. 8d). Unlike its counterparts in the Pacific



and the Atlantic Oceans, the equatorial IO experiences only transient upwelling due to changes in wind direction
associated with migration of the intertropical convergence zone. This also leads to surface currents reversing their
direction four times a year. Thus, the region experiences westward surface currents of weak magnitude during the
southwest and northeast monsoon months and much stronger eastwards current during the spring and fall
intermonsoon months (Han et al., 1999). These narrow eastwards surface currents during the intermonsoon
months, known as Wyrtki jets, are in response to westerly winds (Wyrtki, 1973). The biogeochemical
characteristics of the region have only been recently explored with the help of satellite and *in situ* data (e.g.,
Prasanna Kumar et al., 2012; Strutton et al., 2015). Deepening of the surface layer associated with the eastward
transport of water during the intermonsoon months lowers productivity (Prasanna Kumar et al., 2012).
Chlorophyll concentrations, although much lower compared to the rest of the IO, peaks during October-December
possibly due to wind stirring or shear instability at the base of the eastward moving Wyrtki Jet (Strutton et al.,
2015).  Additionally, *in situ* measurements in the central equatorial IO have revealed deep chlorophyll maxima
located ~60 m depth contributing to more than 30% of the total chlorophyll biomass (Vidya et al., 2013). The
peak ocean DFe concentration is encountered during August-November. Overall, comparison between CTRL,
ATM and SED cases show that atmospheric deposition, peaking during July (Fig. 8d), dominates DFe contribution
to the central equatorial IO, whereas sedimentary DFe plays a distant secondary role (Fig. 11).
Horizontal advection is the most important process of DFe supply within the mixed layer during March-May and
September-November (Fig. 11). During the intervening months, vertical advection plays the predominant role in
DFe supply. Decomposing the horizontal advection further into DFe´ and U´ reveals that the meridional velocity
convergence is the main contributor to the central equatorial IO DFe budget during March-May and September-
November (Fig. S13). This originates from the westerly wind directing equatorward Ekman flow in both the
hemispheres, which leads to convergence and drives eastward propagating downwelling Kelvin wave (McPhaden
et al., 2015). Averaged over the upper 100 m, zonal velocity convergence, although somewhat of lower magnitude,
opposes meridional velocity convergence throughout the year. When the Wyrtki jet weakens, upwelling induced
by easterly wind drives upward vertical supply of DFe, whereas there is downward vertical removal of DFe during
the intervening periods. This alternating between upwelling and downwelling control on DFe has an upward phase
propagation. An important feature of the central equatorial IO, in contrast to other equatorial regions, is the
presence of transient Equatorial Undercurrent between 60 m-200 m depth with core generally centered on the
depth of the 20°C isotherm (Chen et al., 2015). The Equatorial Undercurrent appears most strongly during winter-
spring months and with much weaker magnitude during summer-fall months (Chen et al., 2015; Schott &
McCreary, 2001). CESM simulation reveals the signature of the upper part of the Equatorial Undercurrent in
influencing DFe budget. This is characterized by the zonal velocity underneath the mixed layer (~80 m depth)
showing strong eastward transport during January-April and a much weaker eastward transport during September-
November. The horizontal convergence of DFe is prominent during the developing phase of the Equatorial
Undercurrent (December-February and June-August), probably, associated with progressive eastward extension
and strengthening of Equatorial Undercurrent from the western IO. These periods of horizontal DFe convergence
are interspersed with vertical DFe convergence. Superimposed on advection, vertical mixing plays an important
role in bringing subsurface DFe to the surface levels in the upper 30 m, peaking during July-August.



Biological removal of DFe, almost entirely by small phytoplankton, is conspicuous in the upper 40 m and peaks
during September. This is in line with sediment trap studies over the central equatorial IO where peak biogenic
fluxes are detected during the southwest and fall intermonsoon months and are dominated by foraminifera
carbonate (Ramaswamy and Gaye, 2006). Furthermore, *in situ* water samples have shown that picoplankton,
having size less than 10 µm, consists of more than 90% of the phytoplankton biomass in central equatorial IO
(Vidya et al., 2013). The period of peak biogenic flux is also characterized by peak in DFe removal by scavenging
and remineralization of DFe released from mortality and grazing at deeper layers (Fig. S15c). A secondary
increase in biological removal of DFe is noticed during January-March associated with a secondary peak in
chlorophyll, although its impact is not evident in sediment trap biogenic flux data (Vidya et al., 2013). This might
arise from remineralization of DFe being almost twice the magnitude of scavenging losses during this time of the
year.



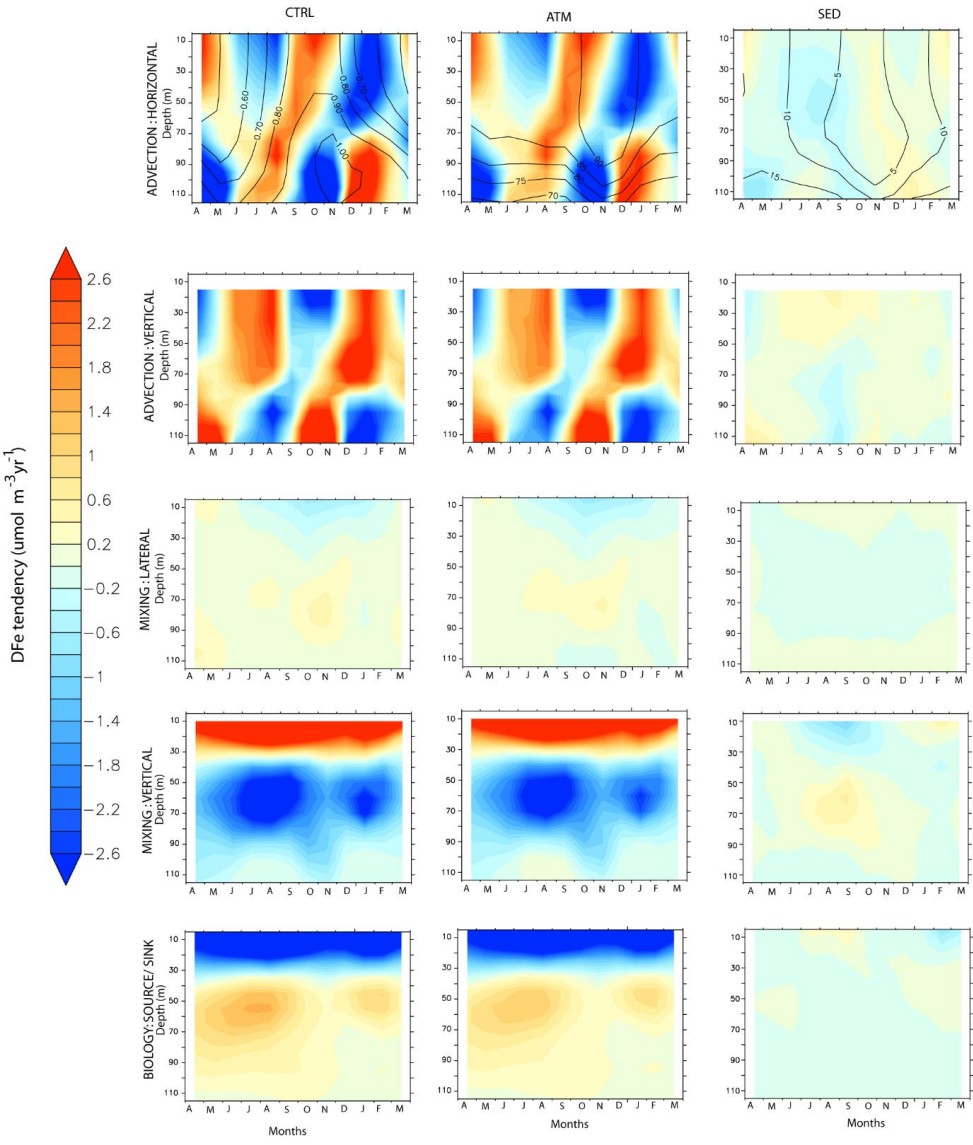

Figure 11: Same as Figure 9, except over the central equatorial Indian Ocean.

### 3.4.4 Central Southern Tropical IO

The central southern tropical IO (13º-17ºS and 72º-76ºE) is located in the transition zone between DFe-poor region of the subtropical IO gyre and DFe-enriched northern IO. Of all the regions considered, this receives the lowest atmospheric DFe (Fig. 8e), resulting in DFe limitation of phytoplankton growth particularly during the boreal summer (Fig. 6). Steady southeasterly winds, prevailing throughout the year, transport dust from Australian sources into this region. Peak in dust deposition is during austral spring and summer associated with strong source



activity (Kok et al., 2021; Yang et al., 2021). A secondary peak in dust deposition during austral winter is possibly
associated with enhanced transport. Northern part of the central southern tropical IO lies on the Seychelles-Chagos
thermocline ridge, which is characterized by doming up of the thermocline due to negative wind stress curl
resulting in Ekman divergence (Vialard et al., 2009). The thermocline progressively deepens towards the sub-
tropical southern IO gyre to the south as wind stress curl changes sign to positive. The westward flowing South
Equatorial Current brings low salinity water and nutrients from the Indonesian region. Satellite observed enhanced
chlorophyll concentration during the boreal (austral) summer (winter) months have been attributed to vertical
diffusion (Kǒne et al., 2009; Lévy et al., 2007). Additionally, westward propagating upwelling/downwelling
Rossby waves arrive in this region following La Nina/El Nino event and play a key role in modulating sea surface
height and the depth of thermocline (Masumoto & Meyers, 1998; Périgaud & Delecluse, 1992). This perturbs the
depth of nitracline, which has significant impact on column productivity (Kawamiya & Oschlies, 2001).
Both ATM and SED sources are important in this region for DFe supply, with the SED (ATM) source having
higher contribution during austral winter (summer) months (Fig. 12). Analysis of CESM-simulated DFe budget
reveals that vertical mixing in the upper 30 m is the most important process of DFe supply, which peaks during
September. This is the time of the year when CESM records the lowest sea surface temperature resulting in mixed
layer deepening. Such winter mixing leads to erosion of vertical gradient in DFe observed during the rest of the
year in the upper 120 m. Horizontal advection is the next most important supplier of DFe in this region. The
westward flowing South Equatorial Current is strongest during austral winter and during winter-to-summer
transition months. This results in meridional velocity convergence and zonal velocity divergence resulting in a
quasi-balance between DFe supply and removal (Fig. S14). Overall, horizontal advection leads to predominantly
sedimentary DFe convergence during March-June and predominantly atmospheric DFe convergence during
September-November.
The wind stress curl is mostly negative, that is upwelling favorable, throughout the year. Between April-October
(austral winter), when winter convection-driven blooms are prominent, wind stress curl becomes weakly negative
to slightly positive. Following this, during January-March, the wind stress curl becomes strongly negative
resulting in upward velocity and favors vertical advection of both atmospheric and sedimentary DFe in equal
magnitude. While vertical U´ is responsible for supplying DFe in the upper 50 m, vertical DFe´ is important at
deeper depths (Fig. S14).
The biological sink of DFe peaks during the month of maximum vertical mixing, that is, during September.
During this time, uptake of DFe is dominated by diatoms, which accounts for more than 80% of the total DFe
uptake. Small phytoplankton dominate the rest of the year. Scavenging removal of DFe and remineralization peaks
one month later during October between 50-90 m depth range (Fig. S15d). Overall, the central southern tropical
IO is the only region where atmospheric deposition and sedimentary sources of iron are equally important in
driving the DFe budget.




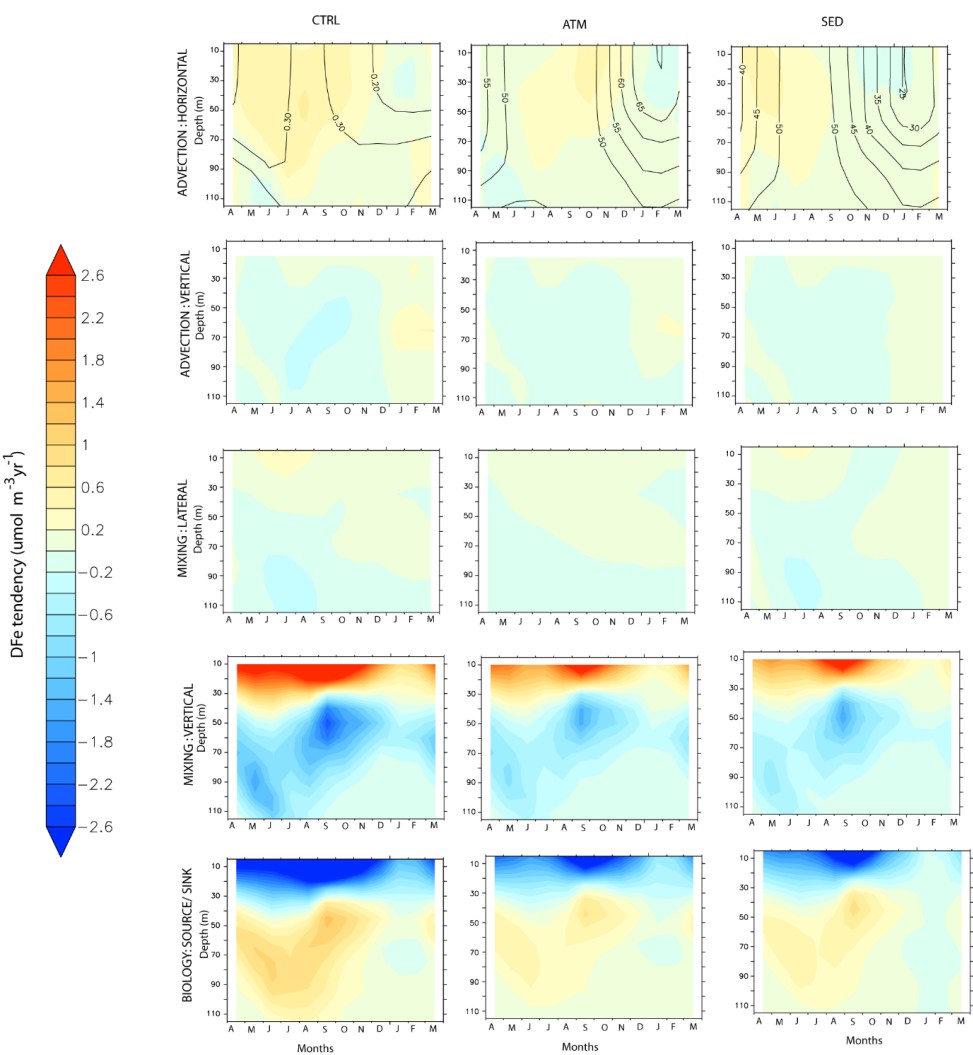


**Figure 12: Same as Figure 9, except over the central southern tropical Indian Ocean.**


**4 Conclusions**

Using the ocean component of the Earth system model CESM version 2.1, this study elucidates the impacts of
various sources of DFe on upper ocean productivity, nutrient limitations and DFe budgets over the northern IO.
The iron cycle in CESM represents the complex interplay between several processes including DFe supply,
removal by scavenging and biological uptake, DFe remineralization, and organic ligand complexation. The major
sources of DFe for this region are included in this model: atmospheric deposition, sediments, hydrothermal vents,
and rivers. Although there are model biases in representing physical and biogeochemical variables, the overall
patterns of spatial and temporal variation of DFe are simulated reasonably well in CESM.



The study finds that atmospheric deposition is the most important source of DFe to the northern IO. Atmospheric
deposition contributes well over 50% of the total DFe concentration and more than 10% (35%) to upper 100 m
(surface level) chlorophyll concentrations, especially over the AS, equatorial IO, and southern tropical IO.
Sedimentary sources become important along continental shelves, where they can contribute to more than 20% of
total DFe. The sedimentary source has the largest impact in fueling phytoplankton blooms over the southern
tropical IO during June-September. In contrast, hydrothermal and river sources have negligible impacts on upper
ocean DFe pools in this region. Almost all regions that experience significant positive chlorophyll responses to
atmospheric as well as sedimentary sources of DFe show a preponderance of diatoms over other phytoplankton
groups. The increases in phytoplankton following external DFe addition are evoked in regions with low
background DFe levels (<0.3 nM) and high initial $NO_3$:DFe, indicating the importance of high levels of
macronutrients. Following, external DFe addition, a shift to nitrate limitation of phytoplankton is observed.
Analysis of DFe budget across different biophysical regimes in the northern IO shows that this budget is generally
dominated by atmospheric deposition, with sedimentary sources of DFe being a distant second contributor. The
exception to this occurs over the southern tropical IO region, where both atmospheric and sedimentary sources
become equally important. In all the regions considered, vertical mixing is the most important physical mechanism
through which DFe is supplied, and furthermore this mechanism is active almost throughout the year. In contrast,
the importance of horizontal and vertical advection is highly seasonal. DFe uptake by small phytoplankton in the
upper ocean is the most important route through which DFe removal takes place, except in the productive waters
where diatoms also participate in the removal process. At subsurface levels, competition between the removal of
DFe by scavenging and its remineralization determines the DFe pool available to the surface ocean via these
aforementioned physical processes.
Of all DFe sources, atmospheric deposition is most likely vulnerable to future global warming, and changes to it
will perhaps exert strong influence on upper ocean productivity and nutrient limitation. This study thus provides
foundations to explore how future scenarios of atmospheric deposition can impact biogeochemistry over the
northern IO.

**Code and data availability**
Climatology of ocean temperature, salinity and nutrients are from World Ocean Atlas 2018 available at
https://www.ncei.noaa.gov/access/world-ocean-atlas-2018/ . Monthly surface chlorophyll data from OC-CCI is
obtained from https://www.oceancolour.org/. Monthly climatology of ocean mixed layer depth based on Holte at
al. (2017) is downloaded from http://mixedlayer.ucsd.edu/. Surface ocean current data from OSCAR can be
downloaded                    from:                 https://podaac.jpl.nasa.gov/dataset/OSCAR_L4_OC_third-
deg?ids=Keywords:Keywords:Projects&values=Oceans::Solid%20Earth::OSCAR&provider=PODAAC.
Dissolved    iron    from   GEOTRACES   Intermediate   Data   Product   2021   is   available   at
https://www.geotraces.org/geotraces-intermediate-data-product-2021/. Additionally, dissolved iron profile data
are also obtained from Tagliabue et al. (2012) available at https://www.bodc.ac.uk/geotraces/data/historical/. The
code for CESM2.1 can be downloaded from https://www.cesm.ucar.edu/models/cesm2/release_download.html
(last access: 01 December 2020).



**Author contributions**
PB conceived the study, carried out model simulations, analysed the data and wrote the manuscript.
**Competing interests**
The author declares that there is no conflict of interest.
**Acknowledgments**
PB acknowledges the computational facilities provided by Supercomputer Education and Research Centre
(SERC) at the Indian Institute of Science for carrying out CESM simulations.
**Financial support**
The author is supported by Department of Science and Technology INSPIRE Faculty scheme
(DST/INSPIRE/04/2018/002625).

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
