# Peer review of "Importance of multiple sources of iron for the upper ocean"

_Biogeosciences, 2022_

## Author Response (AR1)

**RESPONSE TO REVIEWER 1**

I thank the reviewer, Dr. Anh Pham, for the insightful comments and suggestions, which have helped me to improve both the present and the previous version of the manuscript. I have provided responses to the comments; the reviewer comments **(RC)** appear as normal font, my response (**AR)** in *italics* below the respective comments and I have used *blue italics* to quote the changes in the revised manuscript.

**RC:** In this article, Dr. Banerjee performed a suite of computer simulations in a relatively complex ocean biogeochemistry model, which includes a state-of-the-art ocean iron (Fe) cycling scheme, to quantify for the relative roles of different sources of dissolved Fe (dFe) on controlling the dFe budget, primary productivity, phytoplankton composition, and nutrient limitation in the northern Indian Ocean (IO). By comparing results of different simulations in which a certain external source of dFe is removed with results of a simulation in which all dFe sources are considered, the author showed that atmospheric deposition is the most important source of dFe to the dFe budget and phytoplankton growth in the upper northern IO. Sedimentary dFe release plays a secondary role and is locally important near the continental shelves and in the southern tropical IO, while the impact of dFe fluxes from hydrothermal vents and river discharges on the upper northern IO biogeochemistry is negligible. More importantly, by analyzing the nutrient limitation status in the northern IO through these model simulations, the author suggested that phytoplankton growth is most sensitive to external sources of dFe in regions where the background dFe concentration is low and the nitrate-to-Fe ratio is high. In those regions, the increase in phytoplankton growth when additional source of dFe is considered is driven mostly by an increase in diatoms. Finally, by analyzing the dFe budget over five biophysical regimes in the northern IO (the western Arabian Sea, the northern Arabian Sea, the southern Bay of Bengal, the central Equatorial IO, and the central southern tropical IO), the author demonstrated that in the surface ocean, vertical mixing is the most important physical mechanism supplying dFe throughout the year. At the subsurface levels, the dFe budget is balanced through scavenging and remineralization processes.

In my opinion, these results while not surprising are still important for understanding ocean Fe cycling in an important ocean region where primary production is high, biogeochemical cycles of various chemical elements are linked, the ocean circulation is highly dynamic, and both biogeochemical and physical processes are sensitive to climate variabilities and global warming. I also find the analysis of the dFe budget over different biophysical regimes insightful. Besides, the manuscript is well-written and easy to follow.

As a reviewer of the previous version of this manuscript, I am also happy that my suggestions are taken into consideration and thoroughly addressed by the author in this version. I am happy to endorse its publication with a few questions/comments for clarification below:

**AR:** *I greatly appreciate the positive evaluation of the manuscript. My responses to the specific comments follow.*

**RC:** Lines 127-129: How are iron and other tracers initialized in the model?

**AR:** *Ecosystem tracers, including iron, chlorophyll, dissolved organic/inorganic carbon are initialized from a previous simulation using CESM1. Temperature and salinity are initialized from January-mean*

*values obtained from the Polar Science Center Hydrographic Climatology (Levitus et al., 1998). This information is included in the revised version of the manuscript (L131-134).*

**RC:** Lines 147-148: I am surprised that the Fe fluxes from river are quite small even though the author assumed a high constant concentration of dFe in rivers (10nM - line 166)

**AR:** *This is an interesting point indeed. The impact of dissolved Fe (DFe) from river is mostly concentrated in the fresher water within the upper 30 m of the water column to the north of $21^oN$ over the Bay of Bengal. Within this limited region surface concentration of DFe from river exceeds 1 nM throughout the year.  The contribution of river-DFe to total surface DFe is ~60% near $23^oN$, but quickly reduces to ~10% in the vicinity of $20^oN$ latitude. This is probably related to quick scavenging loss of DFe from river. Additionally, there is a strong seasonality to river discharge, leading to river-derived DFe peaking during March and November. Together, this leads to an overall low contribution of river DFe to the open ocean. I have added the following sentence in the revised version of the manuscript to make this clear: "River sources contribute negligibly to total DFe concentrations (Fig. 4d), except in the immediate vicinity of the mouths of large river systems in the northeast BoB: the Ganges-Brahmaputra and the Irrawady-Sittang-Salween. This can arise from the fact that DFe from river is mostly concentrated within the fresher upper 30 m of the water column to the north of $21^oN$ over the BoB and also due to high scavenging losses of iron at the river mouth." Please refer to L469-472 of the revised manuscript.*

*For the reference of the reviewer, I am also providing a zoomed diagram of DFe concentration and scavenging from river source along $91^oE$ longitude, which shows how DFe from river is quickly lost near the river mouth.*

[Figure]

**Fig. 1. (a)** *Shading shows DFe concentration only from river source along 91ºE longitude averaged for last 10 years of CESM simulations. The black contours are the percentage contribution of DFe from rivers to total DFe concentration.* **(b)** *Scavenging loss of DFe from river source along 91ºE longitude.* **(c)** *Percentage contribution of DFe from river to surface (black curve) and 0-100m averaged (red curve) DFe concentration along 91ºE.* **(d)** *Seasonal cycle of DFe concentration from river averaged over 21-23ºN latitude and 90-93ºE longitude.*

**RC:** Lines 161-163: What are the constant low background fluxes here? What is its value?

**AR:** *The constant low background flux can be obtained from CESM-MARBL Github code repository (*[https://github.com/marbl-ecosys/marbl-forcing/](https://github.com/marbl-ecosys/marbl-forcing/)*) as:*

*fesedflux_oxic (µmol m$^{-2}$ d$^{-1}$) = coef_fesedflux_current_speed2 X sedfrac_mod X current_speed$^2$*

*where, coef_fesedflux_current_speed2 = 0.0006568 * 1.2*
*        sed_frac_mod= fraction of each cell that is ocean bottom at each depth*
*        current speed = bottom current speed*

*However, I would like refrain from placing this in the main manuscript until I come across publications from the model developers giving more information on how this coefficient has been derived.*

**RC:** Line 177: What is the value for the constant desorption rate?

**AR:** *The value for the constant desorption rate for scavenged Fe from particles is 1.0 X 10$^{-6}$ cm$^{-1}$. This is now included in L194 of the revised version of the manuscript.*

**RC:** Lines 259-260: Is this underestimation an implication that iron limitation here is not strong enough in the model?

**AR:** *Yes, I agree that underestimation of nitrate along with overestimation of DFe likely leads to, in general, weaker iron limitation in the model than in observations. I am including a line in the revised manuscript to indicate this (L418-420): "To summarize, the ocean component of CESM has deeper MLD than observations, underestimates nitrate and chlorophyll, and overestimates DFe concentrations. Together, this can result in weaker iron-limitation in the simulations compared to observations."*

**RC:** Lines 831-834: I think Fe release from low-oxygen sediments is also vulnerable to global warming since the ocean oxygen level is a function of many biogeochemical and physical processes which are bound to change.

**AR:** *This is a good point and I agree with the reviewer. I am including the following sentences in the "Conclusion" section of the revised manuscript to indicate the importance of iron from low-oxygenated sediments:" Additionally, 59% of the continental shelves and bathyal sea floor over the northern IO experiences hypoxic conditions (Helly and Levin, 2004) and there are several lines of evidence pointing to reductions in oxygen content over this region during the last few decades due to enhanced upper ocean stratification (Schmidtko et al., 2017). This will possibly impact the flux of iron from reduced sediments. The present study thus provides foundations to explore how different future scenarios of atmospheric deposition and the extent of reducing sediments can impact biogeochemistry over the northern IO." Please refer to L937-943 of the revised manuscript.*

Anh Pham

**Additional references**

Helly, J. J. and Levin, L. A.: Global distribution of naturally occurring marine hypoxia on continental margins, Deep-Sea Res. Pt. I, 51, 1159–1168, 2004.

Levitus, S., T. Boyer, M. Concright, D. Johnson, T. O'Brien, J. Antonov, C. Stephens, *and* R. Garfield: Introduction, Vol. I, World Ocean Database 1998, *NOAA Atlas NESDIS 18, 346 pp, 1998.*

Schmidtko, S., Stramma, L., and Visbeck, M.: Decline in global oceanic oxygen content during the past five decades, Nature, 542, 335–339, https://doi.org/10.1038/nature21399, 2017.

**RESPONSE TO REVIEWER 2**

I thank the reviewer, Dr. Nicola Wiseman, for the constructive comments and suggestions, which have helped me to improve the manuscript. I have provided responses to the comments; the reviewer comments (**RC**) appear as normal font, my response (**AR**) in *italics* below the respective comments and I have used *blue italics* to quote the changes in the revised manuscript.

**RC:** In this article, Dr. Banerjee utilized the CESM ocean and marine ecosystem model components to investigate the contributions of various iron sources to the Indian Ocean. This model is well suited for the study due to its complex iron cycle representation and robust ecosystem parameterization. The author specifically investigated the relative contributions of each soluble iron source to the total dissolved iron budget as well as biological productivity on a seasonal basis. The author concludes that atmospheric iron is the primary contributor to the dissolved iron budget and fuels productivity in much of the Indian Ocean, while sedimentary iron follows second, and has impactful contributions in continental shelf regions, as well as where dust deposition is at its minimum. This study clearly defines the role of each iron source to biological productivity and concludes by highlighting the uncertainty of atmospheric iron deposition in a changing climate.

Overall, the author performed a well through out series of experiments that clearly defines the interactions between iron supply and physical drivers in multiple regions of the Indian Ocean. I endorse this paper for publication with the following minor questions/comments for clarification below:

**AR:** *Many thanks for endorsing the paper for publication. My responses to the specific comments are provided below.*

**RC:** Lines 197-199: You mention that freshwater fluxes are calculated from monthly stream flow observations and CLM model. Do you mean from CLM5? What specific output from CLM5, if that is what you are referring to, are you using to derive freshwater fluxers?

**AR:** *The input file for monthly streamflow is based on the river-based estimates of continental freshwater discharge which was originally produced by Dai and Trenberth (2002) based on Bodo et al. (2001) and later extended by Dai et al. (2009). The data contains monthly station-based streamflow for the world's 925 largest rivers and is supplemented by data from several sources as outlined in Dai et al. (2009). Additionally, missing data in the resultant streamflow time-series have been filled with streamflow simulated by CLM3. For this, a linear regression equation has been employed for each river with the CLM-simulated flow as input to obtain streamflow estimates for years without observations. This has been clarified in L217-219 of the revised version of the manuscript as: "Monthly streamflow since 1948 used in this study has been previously derived from gauge data, where linear regression was also employed using CLM3 model streamflow to fill-in missing data (Dai et al., 2009)."*

**RC:** Lines 293-295: What type of correlation coefficient are you utilizing here? It is the Pearson product-moment correlation coefficient or a rank correlation? How are you calculating this statistic?

**AR:** *Pearson product-moment correlation coefficient is used here. Significance of correlation is calculated using Student's t-test with n-2 degrees of freedom, where n is the sample size. Please note that I have also included data from cruise GI-03 in the revised manuscript, which has resulted in*

*improved correlations between observed and simulated DFe concentrations. This information has now been included in L315-319 of the revised manuscript.*

**RC:** Lines 378-379: Do you have maps showing the iron inputs for each field (atmospheric (with black carbon separated), sedimentary, river, vent)? While Fig. 3 shows the contribution to the total DFe averaged over the upper 100m, a supplementary figure with of each input would strengthen the conclusions made regarding the spatial distributions of the sources in the first paragraph of section 3.2.

**AR:** *Thank you for this suggestion. I have now included a figure showing iron input fluxes for each field as Supplementary Figure S1. This is also shown below.*

[Figure]

*Fig. S1 Iron fluxes from the various sources considered in CESM-MARBL over the northern Indian Ocean. Contours in (a) show the fractional contribution of black carbon to atmospheric iron flux.*

**RC:** Lines 528-534: Cellular Fe:C ratios are reported as Fe:C, not DFe:C. Diatom observations have also been expanded since de Baar et al., 2008 and can be greater than 2.00 x 10-4.

**AR:** *Thanks for pointing this out! I have corrected DFe:C to Fe:C in the revised manuscript. I have also included the suggested citations and have recalculated N:Fe ratios based on the suggested citations. This is reflected in the revised manuscript as follows (L632-636):*

*"Over the world oceans, a wide range of cellular Fe:C ratios has been observed for diatoms, ranging from 100 μmol mol⁻¹ for DFe-replete conditions (Twining et al., 2015; 2021) to 2 μmol mol⁻¹ for DFe-deplete conditions (de Baar et al., 2008). Assuming a C:N ratio of 117:16 (Anderson and Sarmiento,*

*1994), the range of N:Fe ratios obtained are ~1000 and ~68000, respectively, for DFe-replete and DFe-deplete conditions."*

Suggested citations: Twining BS, Rauschenberg S, Morton PL, Vogt S (2015) Metal contents of phytoplankton and labile particulate material in the North Atlantic Ocean. Progress in oceanography, 137:261–283.

Twining BS, Antipova O, Chappell PD, Cohen NR, Jacquot JE, Mann EL, Marchetti A, Ohnemus DC, Rauschenberg S, Tagliabue A (2021) Taxonomic and nutrient controls on phytoplankton iron quotas in the ocean. Limnology and oceanography letters, 6(2):96–106.

**Additional references**

Bodo, B. A.: Annotations for monthly discharge data for world rivers (excluding former Soviet Union). NCAR Rep., 111 pp., 2001.

Dai, A., and K. E. Trenberth: Estimates of freshwater discharge from continents: Latitudinal and seasonal variations, *J. Hydrometeor.*, **3,** 660–687, 2002.

Twining, B. S., Rauschenberg, S., Morton, P. L., and Vogt, S.: Metal contents of phytoplankton and labile particulate material in the North Atlantic Ocean, Progr. Oceanogr., 137, 261–283, https://doi.org/10.1016/j.pocean.2015.07.001, 2015.

Twining, B. S., Antipova, O., Chappell, P. D., Cohen, N. R., Jacquot, J. E., Mann, E. L., et al.: Taxonomic and nutrient controls on phytoplankton iron quotas in the ocean, Limnology and Oceanography Letters, 6(2), 96–106, https://doi.org/10.1002/lol2.10179, 2021.

**RESPONSE TO REVIEWER 3**

I would like to thank the reviewer for the valuable feedbacks and helpful suggestions, which have helped to improve the work. I have addressed all the issues raised by the reviewer as detailed below. The reviewer comments **(RC)** appear as normal font, my response **(AR)** in *italics* below the respective comments and I have used *blue italics* to quote the changes in the revised manuscript.

**RC:** The author investigated the impact of different external iron sources into the northern Indian Ocean on phytoplankton growth using CESM. A control simulation was first presented considering four external iron sources: dust, sediments, hydrothermal vents and rivers. Then a series of sensitivity experiments were conducted with one of the four sources set to zero. The differences to the control simulation were used to illustrate contributions of single sources to surface DFe and chlorophyll distributions. At the end, mechanisms of DFe supply in defined biophysical regimes in this region were discussed.

The study area is important in the marine iron and carbon cycle due to high iron input and high biological productivity. The manuscript has a clear structure, and the experiments were designed and conducted in a reasonable way. However, I have some major concerns that some details of the iron model are not clearly and concisely described, and the discussion of model results not always supported by rigorous reasoning. Below are my general comments.

**AR:** *Based on the suggestion by the reviewer I am including several modifications to the revised version of the manuscript. These are summarized below:*

*(i)     Clarifying the points raised by the reviewer with respect to the results of previous studies on ocean iron in the "Introduction" section.*
*(ii)    Addition of more detail in the description of CESM iron cycle along with a schematic diagram.*
*(iii)   A detailed analysis of bias in simulated dissolved iron (DFe), attribution of bias to the source strength over the Arabian Sea (AS) and the Bay of Bengal (BoB), and a discussion of the implications of these biases for the overall conclusions of the study.*
*(iv)    A discussion on what leads to shifts in phytoplankton species composition in CESM following external iron addition.*

**RC:** General comments:

1. This is a modelling study on the iron cycle. A comprehensive understanding of the marine iron cycle and a precise and detailed description of the modelled iron cycle are required to analyse the model results and also to convince readers. The introduction of previous studies in 'Introduction' is not very precise. Here I give two examples:

L33-35: the author stated that several iron addition experiments demonstrated its significance in CO2 drawdown. In fact, iron addition experiments hardly demonstrate a significant effect on CO2 drawdown, since only one of the ship experiments detected a significant increase in carbon export and the others only observed chlorophyll increase induced by iron addition which is not necessarily relevant to CO2 drawdown. This is nicely summarised in Yoon et al. (2018). And the citations in L34-35 are for both natural and artificial fertilisation and do not fit the sentence.

**AR:** *I agree with the reviewer that the sentence in L34-35 does not correctly capture the results from iron fertilization experiments. In L33-36 of the revised version of the manuscript, I am changing this sentence as follows:*

*"Several artificial iron addition experiments performed in the open oceans have demonstrated its significance in regulating phytoplankton growth (Yoon et al., 2018), while natural iron fertilizations have also shown high levels of carbon export from the upper ocean following increased productivity (e.g., Blain et al., 2007; Pollard et al., 2009)."*

**RC:** L45-47: the author stated that hydrothermal vents can only impact productivity where these vents are located at shallow depths. This is not necessarily true. Considering mechanisms to stabilise iron released from hydrothermal vents, this iron could be transported far from vents and upwelled to the surface. And this is not something really new. Papers published 10 years ago already discussed different mechanisms preventing the precipitation of iron in near-vent fields (e.g. Sander and Koschinsky, 2011; Yücel et al. 2011).

**AR:** *Thanks for pointing this out. I am modifying the sentence in L46-48 as: "This is because while atmospheric and sedimentary DFe can impact productivity over both the open and coastal oceans, iron from hydrothermal vents reaching the surface water depends on deepwater ventilation and stabilizing impact of organic ligands (Tagliabue et al., 2010; Sander and Koschinsky, 2011)."*

**RC:** Further in the 'Data and model' chapter, the description of the iron model (L169-183) does not have a clear structure and sometimes confusing. Readers need to know how many iron pools are considered in the model, which processes transfer iron between these pools and how these processes are described as equations. And the first two points are better shown in a scheme. If the code of the iron model was not changed for this study, previous model descriptions can be referred but a brief summary with the main features is still needed for understanding this manuscript without reading another one. If something was changed in the code for this study, please underline and explain these changes and give the equations. This part of model description is central for the manuscript and therefore I expected a much higher quality here.

**AR:** *Based on the suggestion by the reviewer, I have now made several changes to the description of the iron cycle in CESM to make the structure clearer. I have added more details to explain the processes that impact the dissolved iron pool in CESM. I have also included a schematic diagram representing the iron cycle in CESM. Please note that all subsequent figure numbers have changed in the revised manuscript. The changes incorporated are shown below (L177-198 of the revised manuscript):*

*"Iron input to the ocean is balanced by losses from biological uptake and scavenging. The biological uptake of iron is based on the species-specific Fe:C ratio, which varies based on ambient DFe concentration, as discussed previously. The biological uptake term also includes routing of phytoplankton iron to zooplankton based on its feeding preference. Losses of iron from the biological pools are through mortality, aggregation, grazing upon phytoplankton by zooplankton, as well as higher trophic grazing on zooplankton (Long et al., 2021). The scavenging loss of DFe is expressed as a two-step process similar to the thorium scavenging model: involving the calculation of the net adsorption rate to sinking particles and modification of this rate by the ambient iron concentration (Moore and Braucher, 2008). The total sinking particles consist of particulate organic carbon, biogenic silica, calcium carbonate, and dust, which strongly influence DFe scavenging in excess of ligand concentrations. The particulate organic carbon is multiplied by 6 to account for the non-carbon portion*

*of the organic matter that can take part in scavenging. In CESM, scavenging increases non-linearly with DFe concentration. About 90% of the scavenged iron enters the sinking particulate pool, while the rest is lost to sediments. Along with the scavenging contribution, iron released from grazing and mortality of autotrophs and zooplankton also enters the particulate iron pool. Remineralization of this sinking particulate iron replenishes DFe and is parameterized as a function of sinking particulate organic carbon flux. This results in maximum remineralization taking place within the upper 100 m where particulate organic carbon flux is the highest. Additionally, slow desorption of sinking particulate iron also releases DFe at depths and is calculated using a constant desorption rate of 1.0 X 10$^{-6}$ cm$^{-1}$ for particulate iron. The model also includes an explicit ligand tracer for complexing Fe, with ligand sources being from particulate organic carbon remineralization and dissolved organic matter production. Ligand sinks involve scavenging, uptake by phytoplankton, ultraviolet radiation, and bacterial uptake or degradation (Long et al., 2021). An overview of the different sources and sinks of DFe used in CESM-MARL is given in Figure 1."*

[Figure]

*Figure 1: Schematic representation of iron cycle in the ocean component of the CESM model. The texts/boxes/arrows in black show the main processes affecting the dissolved iron pool, while those in red further show what controls the processes impacting the dissolved iron pool. POC (DOC): particulate (dissolved) organic carbon, bSi: biogenic silica.*

**RC:** 2. Based on the arguments provided in the current version, I am not convinced that the control simulation is 'good' enough to serve as a reference for further sensitivity experiments. Figure 2 shows that the model overestimates surface DFe from Dec to May and in subsurface waters along the two transects, particularly the CLIVAR. In L299-350 the author mentioned several potential causes for

overestimation in the subsurface waters: source strength, O2 and ligand concentration, biological uptake and scavenging. Although none of these seems to be able to explain the bias, the author claimed that the result of this simulation 'gives confidence in using the model to study the iron cycle over the region'. In my opinion, there is still much work to do before coming to this conclusion:

**AR:** *Based on the suggestion by the reviewer I have now carried out detailed analysis of the sources of bias in DFe simulated by CESM. These are explained in response to the subsequent comments.*

**RC:** 1) The assumed source strength is of particular importance for this study, since the study aims to quantify contributions of different iron sources in regulating biology. All the sensitivity experiments were made based on this control run. If the control run shows a significant model-data mismatch and the assumed source strength probably causes this bias, more experiments need to done by changing strength of different sources or more analysis of model results, to exclude this possibility. Otherwise, how can the contributions of different sources be examined based on a 'wrong' assumption of source strength? So far a detailed analysis was presented in the manuscript for dust deposition, but not for the other sources. Just saying that it is difficult to exclude the effect of other sources does not sound convincing.

**AR:** *I have now carried out more detailed analysis on the distribution of bias and the possible factors contributing to the bias in CESM simulated iron concentrations. Please note that I have also included data from cruise GI-03 in revised manuscript, which has resulted in improved correlations between observed and simulated DFe concentrations (L316). The main points that emerge regarding the distribution of the bias are: (1) CESM shows a very low magnitude of negative DFe bias to the west of $60^oE$ longitude over the AS, (2) the magnitude of the positive bias is much lower to the south of $5^oS$ latitude compared to the north and (3) Coastal and open oceans also experience similar magnitudes of positive DFe bias, implying that DFe bias might be stemming from multiple sources in the entire domain. This information is now presented as Supplementary Table 1.*

**Table S1.** Magnitude of bias in CESM simulated DFe concentration (nM)

| Region | Bias |
| --- | --- |
| **5S-20$^o$S latitude** | 0.16 |
| **5S-30$^o$N latitude** | 0.47 |
| **Arabian Sea (west of 60$^o$E longitude)** | -0.07 |
| **Open ocean (depth > 1000 m)** | 0.38 |
| **Coastal regions (depth < 1000 m)** | 0.32 |

*On analyzing the possible causes of bias, it is revealed that although the magnitude of dust deposition from CESM is on the lower side compared to observations, the percentage solubility of iron from dust is overestimated by the model compared to observations. Thus, to the west of $60^oE$ longitude over the AS, the underestimation in dust deposition is counterbalanced by overestimation in iron solubility to yield low magnitude of negative bias in simulated DFe concentration. To the east of $60^oE$ longitude, over the AS, DFe is overestimated due to high iron solubility coupled with low magnitude of underestimation of dust deposition. Over the BoB, however, both dust deposition and percentage solubility of iron is underestimated. So, the overestimation of DFe concentration over the BoB does not arise from atmospheric deposition source. On comparing CTRL simulation with NATM and NSED*

*along the CLIVAR transect, considerable contribution of sedimentary sources of DFe, especially, at depths greater than 60 m is seen (in the newly included Figure S6). Furthermore, comparing changes in salinity along with DFe concentration from NATM and NSED case reveals possible enhanced transport of iron from continental shelves leading to positive bias over the eastern IO. I am including the following text in the revised version of the manuscript:*

*L346:386: "Thus, overall, there is some underestimation of dust deposition over the northern IO, which might not explain positive DFe bias in CESM simulations. However, there is a possibility of fractional solubility of Fe from dust having an impact on DFe derived from atmospheric sources. Over the AS, percentage solubility of aerosol has been reported to vary between 0.02 and 0.43% (Srinivas et al., 2012). Considering that Fe constitutes 3.5% of dust by weight and using 0.02% and 0.5% as the lower and upper bounds to Fe solubility, the total fluxes of soluble Fe based on CESM dust deposition are calculated. The calculated iron flux ranges from 0.002 (0.04) $\mu mol\ m^{-2}\ d^{-1}$ over the western AS to 0.01 (0.35) $\mu mol\ m^{-2}\ d^{-1}$ over the eastern AS for 0.02% (0.5%) solubility. The corresponding ranges of soluble Fe flux from CESM is 0.05 $\mu mol\ m^{-2}\ d^{-1}$ in the west to 0.8 $\mu mol\ m^{-2}\ d^{-1}$ in the eastern AS. Again, using median dust deposition values from DIRTMAP data and assuming 0.5% iron solubility, soluble Fe fluxes vary from 0.12 to 0.17 $\mu mol\ m^{-2}\ d^{-1}$ from west to east AS. It is therefore clear that CESM model input of soluble Fe from atmosphere is overestimated compared to observations. This inference does not change even after adding the contribution of black carbon (after assuming 6% solubility of Fe) to the atmospheric iron flux. This is because fractional solubility of Fe in CESM varies from 1.2% over northwestern AS to ~5% over the southern AS. Ship-based measurements, on the other hand, have observed that high levels of $CaCO_3$ in the dust over the AS acts as a neutralizing agent, leading to much lower aerosol solubility (Srinivas et al., 2012). Additionally, for the GI05 transect (Fig. 3g), DFe concentration reduces drastically in the NATM case (Fig. S6 a-c), indicating that dust deposition and its solubility is the major factor contributing to the simulated levels of DFe and its biases.*

*The impact of dust solubility on DFe concentration, however, does not explain the positive biases in simulated DFe over the BoB. The percentage solubility of aerosol iron measured over the BoB is high, varying between 2.3% and 24%, due to presence of acid species from anthropogenic activities (Srinivas et al., 2012). This leads to much higher soluble iron deposition than that is obtained from CESM. For example, in CESM the soluble Fe flux over BoB varies from ~0.05 to 0.35 $\mu mol\ m^{-2}\ d^{-1}$, whereas, calculated soluble Fe flux varies from 0.06 to above 1 $\mu mol\ m^{-2}\ d^{-1}$. Thus, atmospheric supply of iron is possibly underestimated over the BoB. It is, therefore, quite possible that this positive bias in DFe stems from either sedimentary or river sources. In fact, comparing CTRL simulation with NATM and NSED along the CLIVAR transect in Figure 3f, reveals considerable contribution of sedimentary sources of DFe, especially at depth greater than 60 m (Fig. S6 d-f). Furthermore, the latitudinal change in salinity along this transect closely follows the latitudinal pattern of change in DFe from NATM case, but not DFe from NSED case. To examine this, DFe from NATM and NSED cases and salinity from CTRL case have been taken along the CLIVAR transect from depths greater than 60 m and have been detrended. The correlation between DFe from NATM and salinity is -0.75 indicating that non-atmospheric sources of DFe is associated with fresher water transported from the coastal regions. The corresponding correlation between DFe from NSED and salinity is -0.16 indicating that non-sedimentary sources of DFe has no salinity dependence. The underestimation of atmospheric iron deposition along with salinity-dependence of DFe from the NATM case together indicates that enhanced transport of sediments from continental margins is likely to be the source of DFe bias along the CLIVAR transect. One possible explanation is that the low resolution of the model is unable to capture the high velocity of the coastal currents that may limit the spreading of sediments from the coastal regions to the open oceans. The simulated coastal current is weaker than OSCAR observations during April, when the*

*CLIVAR measurements were undertaken (Fig. S6 g-h). This can lead to greater diffusive spreading of iron from the coast into the open ocean. Such an effect of model resolution has been previously shown to result in a higher sedimentary contribution to DFe off the northwest Pacific and southwest Atlantic ocean (Harrison et al., 2018)."*

*L418:425: "To summarize, the ocean component of CESM has deeper MLD than observations, underestimates nitrate and chlorophyll, and overestimates DFe concentrations. Together, this can result in weaker iron-limitation in the simulations compared to observations. Over the AS, the positive bias in simulated DFe is present mostly to the east of 60ºE longitude and can be related to the higher solubility of atmospheric iron in CESM compared to the observations. Over the BoB, DFe bias likely originates from enhanced transport of sedimentary iron from continental shelf margins. To the west of 60ºE, simulated DFe has negative bias of low magnitude, possibly because underestimation of dust deposition is counterbalanced by overestimation of iron-solubility. Over the southern tropical IO, the magnitude of bias is also low compared to the rest of the study domain."*

[Figure]

***Fig. S6** Vertical variation in DFe along **(a-c)** GI05 and **(d-f)** CLIVAR transects for CTRL, NATM and NSED cases. **(g-h)** Surface currents over the Bay of Bengal for the month of April, from (g) CESM simulations and (h) OSCAR retrievals.*

**RC:** 2) Biological uptake rather affects the loss of DFe in surface waters and the effect of scavenging onto organic particles is also stronger in the surface than in the subsurface waters due to the vertical gradient of particle concentration. Thus the subsurface bias (below 60m) might not be explained by

these removal processes. A quantitative analysis of the two loss fluxes along the CLIVAR transect could help to conclude their role.

**AR:** *Thanks for this suggestion. In the revised version of the manuscript, I have included a quantitative analysis of relative importance of Fe uptake by phytoplankton and scavenging losses over the study domain. For phytoplankton Fe uptake, I have used a wide range of Fe:C ratios along with particulate organic carbon export fluxes at 100 m calculated from observations of $^{234}$Th fluxes over the northern IO. For scavenging losses, I have used values estimated by Chinni and Singh (2022), which is based on particulate Fe value from the eastern tropical South Pacific and $^{234}$Th fluxes over the AS. These calculations show that Fe uptake by phytoplankton is possibly underestimated over the AS, which can contribute to some overestimation of DFe in the surface water over AS. Over BoB, Fe uptake is within the range of observation-based values. Scavenging removal simulated by CESM is also within the range of estimated values and is possibly not contributing to DFe bias in the model. This is discussed in L396-417 of the revised manuscript as follows:*

*"With respect to loss terms, biases in Fe uptake and scavenging can impact simulated DFe concentrations, especially in the surface waters. To account for Fe uptake by phytoplankton, particulate organic carbon export fluxes at 100 m calculated from $^{234}$Th fluxes have been used in conjunction with Fe:C ratios. Since the cellular Fe:C ratio varies widely depending on external DFe availability and phytoplankton species composition, a lower bound of 6 $\mu$mol mol$^{-1}$ and an upper bound of 50 $\mu$mol mol$^{-1}$ have been considered. The lower bound is based on measurements over the eastern IO (Twining et al., 2019) where oligotrophic conditions are encountered. The upper bound is based on measurements over the tropical North Atlantic where high dust deposition leading to high surface DFe concentration prevails (Twining et al., 2015). Combining Fe:C values with particulate organic carbon export fluxes from JGOFS cruises (Buesseler et al., 1998) yields Fe uptake by phytoplankton varying between ~0.0004 and ~0.0035 $\mu$mol m$^{-3}$ d$^{-1}$ for all seasons over the AS. Phytoplankton Fe uptake from CESM over the AS varies between ~0.0001 and ~0.002 $\mu$mol m$^{-3}$ d$^{-1}$, which are on the lower end of observation-based values. Over the BoB, phytoplankton Fe uptake varies between ~0.00002 and ~0.004 $\mu$mol m$^{-3}$ d$^{-1}$ based on available POC measurements (Anand et al., 2017; 2018). The corresponding ranges of CESM simulated DFe uptake are ~0.0002 to ~0.001 $\mu$mol m$^{-3}$ d$^{-1}$, which is within the range of values calculated from observations. With respect to scavenging losses, based on particulate Fe value from the eastern tropical South Pacific and $^{234}$Th fluxes over the AS, Chinni and Singh (2022) estimated abiotic removal of 0.001-0.005 $\mu$mol m$^{-3}$ d$^{-1}$ for the upper 100 m. In the present simulations, average scavenging removal is ~0.003 $\mu$mol m$^{-3}$ d$^{-1}$ over both the AS and BoB (range: 0.002 to 0.026 $\mu$mol m$^{-3}$ d$^{-1}$) and reduces to less than 0.001 $\mu$mol m$^{-3}$ d$^{-1}$ to the south of the equator. Overall, Fe uptake by phytoplankton is possibly underestimated over the AS, which can contribute to some overestimation of DFe in the surface waters over this region. Over BoB, Fe uptake is within the range of observation-based values. Scavenging removal simulated by CESM is also within the range of observation-based values and is possibly not contributing to DFe bias in CESM."*

**RC:** 3) A clear increase of DFe in the subsurface waters which spreads from the near-coast region to the open ocean is likely caused by an additional input of iron below the surface in the coastal regions, e.g. sediment. I am wondering if the author checked the subsurface DFe along the CLIVAR transect in simulations without sediment (whether DFe is still elevated below 60m), and whether the sediment resuspension plays a role in this region. Another factor might be the dissolution of iron from the 'soft' dust. Even dust deposition could be underestimated, the slow release of iron from sinking dust particles is not taken up by phytoplankton (which is underestimated anyway) and could contribute to an increase of DFe below the surface waters where the biological uptake and concentration of organic particles

become lower. These are just my hypotheses and this kind of open questions needs to be (quantitatively) analysed.

**AR:** *Thank you for this comment. Please see my response to the previous comments, where I have tried to address this.*

**RC:** After the causes of the bias are found, it will be further checked if the causes strongly affect the analysis of source contribution or will affect DFe distribution in a systematical way that the relative differences between runs can still be assigned to different source strengths. Then the author can convince readers that this control run can be used as a reference for further sensitivity experiments. So I am not saying that this run can not be used or must be further tuned, but its validity needs to be better argued.

**AR:** *In the revised version of the manuscript, I am including a discussion on how the different sources of bias to CESM-simulated DFe can impact the conclusions in the present study at the end of Section 3.2. Calculations performed by including and excluding the regions showing biases arising from a specific source of DFe shows that while biases in the source strength might regionally impact the percentage contribution of DFe from a particular source, the overall conclusion of atmospheric source being the most important for upper ocean DFe over the northern IO, followed by sedimentary source, does not change. This is given below and included in L486-498 of the revised manuscript:*

*"Based on the analysis of origin of bias in simulated DFe concentrations in Section 3.1, it is likely that contribution of atmospheric sources to upper 100 m DFe concentration is overestimated over the eastern AS and the contribution of sedimentary sources to upper 100 m DFe concentration is overestimated over the BoB. Averaging over the entire domain, atmospheric source contributes ~67% to the upper 100 m DFe concentration. On masking out the region to the east of 65ºE longitude over the AS, where the highest positive bias of DFe from dust has been noted, it is seen that atmospheric source contributes ~65% to the upper 100 m DFe concentration. Again, averaging over the study domain, sedimentary source contributes ~30% to the upper 100 m DFe concentration. On masking out BoB, where positive bias of DFe from sedimentary sources has been identified in Section 3.1, it is seen that sedimentary source contributes ~33% to the upper 100 m DFe concentration. Thus, while biases in the source strength might regionally impact the percentage contribution of DFe from various sources to the northern IO, the overall conclusion of atmospheric source being the most important for upper ocean DFe over the northern IO, followed by sedimentary sources, does not change. "*

*Additionally, with respect to DFe source strength impacting phytoplankton response, the following sentences are being included in L596-605, at the end of Section 3.3.2:*

*"It is important to mention here that DFe bias arising from source strength has low impact on phytoplankton response to a particular source of DFe. This is because the strongest phytoplankton response to a specific DFe source is over the western AS and subtropical southern IO. As noted in Section 3.1, these regions have the least magnitude of DFe bias. For example, averaging over the upper 100 m over the northern IO, atmospheric source contributes ~13% to total chlorophyll concentration. Even after masking out the region to the east of 65ºE longitude over the AS, where the highest positive DFe bias arising from atmospheric Fe has been noted, it is seen that atmospheric source contributes ~13% to the upper 100 m chlorophyll concentration. Similarly, sedimentary sources contribute ~9% to the upper 100 m chlorophyll concentration over the entire northern IO domain. Masking out BoB, where DFe bias is due to enhanced sediment transport, results in sedimentary source contributing ~8% to the upper ocean chlorophyll concentration."*

**RC:** 3. The explanation of the phytoplankton community shift in response to iron input is incomplete and not always true (L451-471). Generally, findings in model results should be explained based on model parameterisations. In the manuscript, the community shift is described, and then some possible reasons based on observations and lab experiments are mentioned. However, what explains a similar phenomenon in the reality or in lab is not necessarily the cause of the model behaviour. For example in L452-457, the author cited de Baar et al. (2005) to support the modelled the outcompete by diatoms and cited Sunda and Huntsmann (1995) to explain it with their large cell size and luxury uptake. These can not directly answer the question: what in the model causes the outcompete? Further, large diatoms should not outcompete small phytoplankton by iron uptake, since the surface:volume ratio matters, not the absolute cell surface, otherwise, they would not more suffer from iron limitation if iron is depleted. And I am wondering how this is taken into account in the model. A careful analysis of changes in growth rate and limitation factor of both species can easily reveal the model parameters determining the community shift. It has been already done in several studies. And here again, it is better to show model equations with parameters (in the main part or supplementary material) to make clear in which processes and parameters diatom and small phytoplankton differ in the model.

**AR:** *Many thanks for this suggestion. I have now explained the phytoplankton community shift simulated in response to iron addition with respect to differences in nutrient uptake rate between different species and grazing pressure. This is incorporated in the revised manuscript as follows (L540-553):*

*"Such differences in species response to external iron addition arise from differences in nutrient uptake between different phytoplankton functional groups in CESM. Phytoplankton growth rate ($\mu_i$) is parameterized as a product of resource-unlimited growth rate ($\mu_{ref}$ in $d^{-1}$) at a reference temperature of 30ºC, and three terms that describe nutrient limitation ($V_i$), temperature dependence ($T_f$) and light availability ($L_i$). This is expressed as:*

$$\mu_i = \mu_{ref} V_i T_f L_i \qquad (1)$$

*The nutrient limitation term for iron, Vi, for a specific phytoplankton group i is expressed as:*

$$V_i^{Fe} = \frac{Fe}{Fe + K_i^{Fe}} \qquad (2)$$

*where Fe is the concentration of iron and $K_i^{Fe}$ is the Fe uptake half-saturation constant for a phytoplankton group. While small phytoplankton have been assigned a value of $3.0 \times 10^{-5}$ mmol $m^{-3}$ for $K_i^{Fe}$, diatoms have been assigned a higher value of $7.0 \times 10^{-5}$ mmol $m^{-3}$. This leads to the small phytoplankton outcompeting diatoms when nutrient levels are low. Additionally, small phytoplankton are subjected to higher grazing pressure than diatoms. The maximum grazing rate assigned in CESM is 3.3 $d^{-1}$ for small phytoplankton versus 3.15 $d^{-1}$ for diatoms. Together, the differences in nutrient uptake half-saturation constant and grazing pressure between different phytoplankton species results in diatom dominating blooms under nutrient-replete conditions."*

At this stage I don't think giving more specific comments would help. I just like to encourage the author to improve the model description, do more detailed analysis of model results and support conclusions through better reasoning.

References:

Sander, S., Koschinsky, A. Metal flux from hydrothermal vents increased by organic complexation. Nature Geosci 4, 145–150 (2011). https://doi.org/10.1038/ngeo1088

Yoon, J.-E., Yoo, K.-C., Macdonald, A. M., Yoon, H.-I., Park, K.-T., Yang, E. J., Kim, H.-C., Lee, J. I., Lee, M. K., Jung, J., Park, J., Lee, J., Kim, S., Kim, S.-S., Kim, K., and Kim, I.-N.: Reviews and syntheses: Ocean iron fertilization experiments – past, present, and future looking to a future Korean Iron Fertilization Experiment in the Southern Ocean (KIFES) project, Biogeosciences, 15, 5847–5889, https://doi.org/10.5194/bg-15-5847-2018, 2018.

Yücel, M., Gartman, A., Chan, C. et al. Hydrothermal vents as a kinetically stable source of iron-sulphide-bearing nanoparticles to the ocean. Nature Geosci 4, 367–371 (2011). https://doi.org/10.1038/ngeo1148

**Additional references**

Anand, S. S., Rengarajan, R., Sarma, V. V. S. S., Sudheer, A. K., Bhushan, R., and Singh, S. K.: Spatial variability of upper ocean POC export in the Bay of Bengal and the Indian Ocean determined using particle-reactive $^{234}$Th, J. Geophys. Res.-Oceans, 122, 3753–3770, https://doi.org/10.1002/2016JC012639, 2017.

Anand, S. S., Rengarajan, R., Shenoy, D., Gauns, M., and Naqvi, S. W. A.: POC export fluxes in the Arabian Sea and the Bay of Bengal: A simultaneous $^{234}$Th$/^{238}$U and $^{210}$Po$/^{210}$Pb study, Mar. Chem., 198, 70–87, https://doi.org/10.1016/j.marchem.2017.11.005, 2018.

Buesseler, K., Ball, L., Andrews, J., Benitez-Nelson, C., Belastock, R., Chai, F., and Chao, Y.: Upper ocean export of particulate organic carbon in the Arabian Sea derived from thorium-234, Deep-Sea Res. Pt. II, 45, 2461–2487, https://doi.org/10.1016/S0967-0645(98)80022-2, 1998.

Harrison, C. S., Long, M. C., Lovenduski, N. S., and Moore, J. K.: Mesoscale Effects on Carbon Export: A Global Perspective, Global Biogeochem. Cy., 32, 680–703, https://doi.org/10.1002/2017GB005751, 2018.

Sander, S. and Koschinsky, A.:Metal flux from hydrothermal vents increased by organic complexation, Nat. Geosci., 4, 145–150, https://doi.org/10.1038/ngeo1088, 2011.

Srinivas, B., Sarin, M. M., and Kumar, A.: Impact of anthropogenic sources on aerosol iron solubility over the Bay of Bengal and the Arabian Sea, Biogeochemistry, 110, 257–268, https://doi.org/10.1007/s10533-011-9680-1, 2012.

Twining, B. S., Rauschenberg, S., Morton, P. L., and Vogt, S.: Metal contents of phytoplankton and labile particulate material in the North Atlantic Ocean, Progr. Oceanogr., 137, 261–283, https://doi.org/10.1016/j.pocean.2015.07.001, 2015.

Yoon, J.-E., Yoo, K.-C., Macdonald, A. M., Yoon, H.-I., Park, K.- T., Yang, E. J., Kim, H.-C., Lee, J. I., Lee, M. K., Jung, J., Park, J., Lee, J., Kim, S., Kim, S.-S., Kim, K., and Kim, I.-N.: Reviews and syntheses: Ocean iron fertilization experiments – past, present, and future looking to a future Korean Iron Fertilization Experiment in the Southern Ocean (KIFES) project, Biogeosciences, 15, 5847–5889, https://doi.org/10.5194/bg-15-5847-2018, 2018.